# Direct binding of CEP85 to STIL ensures robust PLK4 activation and efficient centriole assembly

Yi Liu[1,2], Gagan D. Gupta[1], Deepak D. Barnabas[3], Fikret G. Agircan[1], Shahid Mehmood[4], Di Wu[4], Etienne Coyaud[5], Christopher M. Johnson[3], Stephen H. McLaughlin[3], Antonina Andreeva[3], Stefan M.V. Freund[3], Carol V. Robinson[4], Sally W.T. Cheung[1], Brian Raught[5,6], Laurence Pelletier [1,2] & Mark van Breugel[3]

Centrosomes are required for faithful chromosome segregation during mitosis. They are composed of a centriole pair that recruits and organizes the microtubule-nucleating pericentriolar material. Centriole duplication is tightly controlled in vivo and aberrations in this process are associated with several human diseases, including cancer and microcephaly. Although factors essential for centriole assembly, such as STIL and PLK4, have been identified, the underlying molecular mechanisms that drive this process are incompletely understood. Combining protein proximity mapping with high-resolution structural methods, we identify CEP85 as a centriole duplication factor that directly interacts with STIL through a highly conserved interaction interface involving a previously uncharacterised domain of STIL. Structure-guided mutational analyses in vivo demonstrate that this interaction is essential for efficient centriolar targeting of STIL, PLK4 activation and faithful daughter centriole assembly. Taken together, our results illuminate a molecular mechanism underpinning the spatiotemporal regulation of the early stages of centriole duplication.

[1] Lunenfeld-Tanenbaum Research Institute, Sinai Health System, 600 University Avenue, Toronto, ON M5G 1X5, Canada. [2] Department of Molecular Genetics, University of Toronto, Toronto, ON M5S 1A8, Canada. [3] Medical Research Council – Laboratory of Molecular Biology, Francis Crick Avenue, Cambridge, CB2 0QH, UK. [4] Department of Chemistry, University of Oxford, Oxford, UK. [5] Princess Margaret Cancer Centre, University Health Network, 101 College Street, Toronto, ON M5G 1L7, Canada. [6] Department of Medical Biophysics, University of Toronto, Toronto, ON M5G 1L7, Canada. These authors contributed equally: Laurence Pelletier, Mark van Breugel. Correspondence and requests for materials should be addressed to L.P. (email: pelletier@lunenfeld.ca) or to M. van B. (email: vanbreug@mrc-lmb.cam.ac.uk)

Centrosomes are the primary microtubule-organizing center in metazoan cells with additional functions in other cellular processes, such as cell signalling, motility and the regulation of cell shape and polarity[1–3]. The core of centrosomes consists of a pair of orthogonally arranged centrioles, barrel-shaped structures with a peripheral microtubule array that constitutes their wall. When forming centrosomes, centrioles recruit pericentriolar material (PCM), a microtubule nucleating, proteinaceous matrix[4, 5]. Besides the formation of centrosomes,

centrioles (then referred to as basal bodies) are also essential for templating the formation of cilia and flagella, hair-like cellular projections that serve in signalling and motility[1, 6]. Centriole duplication is tightly regulated to ensure a single round of duplication per cell cycle and dysfunctions in this duplication cycle result in numerical centriole aberrations that can cause human disorders, including cancer, developmental brain diseases and ciliopathies[6–11].

Work across different organisms uncovered five evolutionarily conserved components with essential roles in centriole duplication: PLK4, CEP192, SASS6, STIL and CPAP in humans[12–16]. Following centriole disengagement at the end of mitosis, centriole duplication formally initiates at the G1/S transition when the CEP152 and CEP192 pool, enveloping the mother centriole, cooperate to recruit PLK4 kinase in a Polo-box domain 1 and 2 (PB1 and PB2) dependent manner[17,18]. Subsequently, through its C-terminal PB3, PLK4 interacts with STIL and phosphorylates it, enabling STIL to recruit SASS6 to the site of procentriole assembly[19–23]. SASS6 in turn oligomerises to form a central, ninefold-symmetric scaffold (the centriolar cartwheel), which enforces centriole symmetry and diameter[24–27]. Following cartwheel assembly, CEP135/CPAP is recruited to the periphery of this scaffold by binding to SASS6/STIL and subsequently interact with microtubules to stabilize them[28–30].

In this ordered assembly cascade, the tight regulation of PLK4 activity plays a pivotal role. Misregulation of PLK4 activity levels has drastic consequences for centriole formation and has been linked to cancer progression[8,11]. While low levels of PLK4 activity impair centriole duplication, high levels of PLK4 activity lead to centriole overduplication[31–33]. PLK4 activity is regulated through autophosphorylation of its activation loop which in turn promotes trans-autophosphorylation of regulatory sites creating a phosphodegron that is subsequently bound by the SCF$^{Slimb/\beta-TrCP}$ ubiquitin ligase leading to PLK4 ubiquitylation and destruction[34–38]. Through this autoinhibitory mechanism PLK4 kinase activity is thus tightly controlled to ensure a single round of centriole duplication per cell cycle.

STIL has been implicated as a PLK4 regulator, possibly through relieving PLK4 autoinhibition by interacting with PLK4 PB3 and its Linker 1 (L1) region[22,23,29,38–40]. Consistent with this role, as for PLK4, depletion of STIL prevents centrioles formation, whereas its overexpression triggers centriole overduplication[41,42]. Further underpinning this important function, STIL levels in cells are tightly controlled, with a cell cycle dependent, initial centriolar accumulation at the onset of centriole duplication (at the G1/S transition) and its subsequent degradation during late mitosis in a APC/C dependent manner[41,43].

In the earliest stages of centriolar PLK4 recruitment, PLK4 appears to form a ring around the centriole by interacting with CEP152 already present on the older mother centriole. On the younger mother centriole, PLK4 first interacts with CEP192, forming a ring with an outer diameter of ~440 nm, but after CEP152 recruitment to this centriole, it follows CEP152 to form a wider ring with an outer diameter of ~590 nm[44]. Subsequently and concomitant with the recruitment of STIL and SASS6, the PLK4 ring dissolves, leaving a single dot of PLK4, that co-localizes with STIL and SASS6 at the site of nascent procentriole formation[17,18,22]. The PLK4-STIL-SASS6 complex is maintained at this place during the remaining interphase[41,42,45–47]. It is currently not known whether these events constitute a complete description of the early steps of centriole duplication, or whether there are additional layers of complexities that regulate STIL recruitment and its ability to activate PLK4 to trigger the downstream events of centriole assembly.

Here we identify CEP85 as a regulator of centriole duplication that directly engages STIL. Using structural information of this complex derived from X-ray crystallography, we find that perturbing the CEP85-STIL interaction interface impairs STIL localization to centrosomes, reduces PLK4 kinase activation and consequently impedes procentriole assembly. Together, our findings elucidate the molecular basis behind a previously undescribed modulatory step during the most upstream events of centriole duplication.

## Results

**CEP85 is a regulator of centriole duplication.** To identify factors implicated in centriole biogenesis, we originally sought to map protein interactions of centriole components using proximity-dependent biotin identification (BioID), including the upstream centriole duplication factors STIL, SASS6, CEP152 and CEP63[48]. Since, CEP192 was not initially included in this analysis we submitted it to BioID exactly as performed previously[48] (Supplementary Fig. 1a-c and Supplementary Data 1). Our analysis of the proximity interaction landscape of these five components revealed that CEP85, a centrosomal protein so far only implicated in regulating centrosome disjunction through its interaction with NEK2[49], displayed a prominent proximity signature with a number of centriole duplication factors (Supplementary Fig. 1a). CEP85 was also submitted to BioID analysis and itself displayed interactions with components of this network (Supplementary Fig. 1a-b and 1d). CEP85 was independently identified as a putative interactor of centriole duplication factors[50]. Together these results suggest a potential role for CEP85 in centriole duplication.

We thus decided to investigate CEP85's role in centriole formation (Fig. 1a–f). We depleted endogenous CEP85 in U-2 OS cells using siRNA in cells harbouring a Tet-inducible RNAi-resistant FLAG-CEP85 transgene to rescue loss-of-endogenous CEP85 (Fig. 1c, f). Using CEP192 and Pericentrin (PCNT) as centrosome markers we observed a marked decrease in centrosome numbers in CEP85 RNAi-treated cells relative to the control transfected cells and this effect could be rescued by RNAi-

**Fig. 1** CEP85 is a regulator of centriole duplication. **a–c** Effect of CEP85 depletion on centrosome number. U-2 OS cells expressing Tet-inducible FLAG or the siRNA-resistant FLAG-CEP85 transgene were transfected with control or CEP85 siRNA and induced with tetracycline for 72 h. Scale bar 10 μm, white boxes indicate the magnified region. **b** The graph shows the percentage of cells with the indicated centrosome numbers ($n = 200$/experiment, three independent experiments). **c** Western blot analysis of FLAG-CEP85 protein levels. α-tubulin served as a loading control. **d–f** Impact of CEP85 depletion on centriole number. The G2-phase arrest assays (see Methods) were performed in U-2 OS cells conditionally expressing FLAG or siRNA-resistant FLAG-CEP85, treated with control or CEP85 siRNA and induced with tetracycline for 72 h. Scale bar 10 μm, white boxes indicate the magnified region. **e** Bar graph, the number of centrioles per cell were counted. ($n = 200$/experiment, three independent experiments). **f** Western blot showing the levels of FLAG-CEP85 in control or CEP85 siRNA transfected cells. α-tubulin served as a loading control. **g, h** The role of CEP85 in PLK4-induced centriole overduplication. The PLK4 assays were performed as described in Methods. Scale bar 10 μm, white boxes indicate the magnified region. **h** The graph showing the percentage of cells with over four centrioles ($n = 100$/experiment, three independent experiments). **i, j** The role of CEP85 in S-phase arrest induced centriole overduplication. The assays were performed as described in Methods. Scale bar 10 μm, white boxes indicate the magnified region. **j** The graph indicates the percentage of cells with over four centrioles ($n = 100$/experiment, three independent experiments). **k, l** Determining at which stage CEP85 acts in the centriole duplication pathway using the PLK4-induced centriole overduplication assays (see the Methods). Scale bar 10 μm, white boxes indicate the magnified region. **l** The graph indicates the relative levels of CEP192, Myc-Plk4, STIL, SASS6 and Centrin at centrosomes ($n = 100$/experiment, three independent experiments). Two-tailed $t$-test was performed for all $p$-values, all error bars represent SD, and asterisks for $p$-values are **$p < 0.01$ and *$p < 0.05$

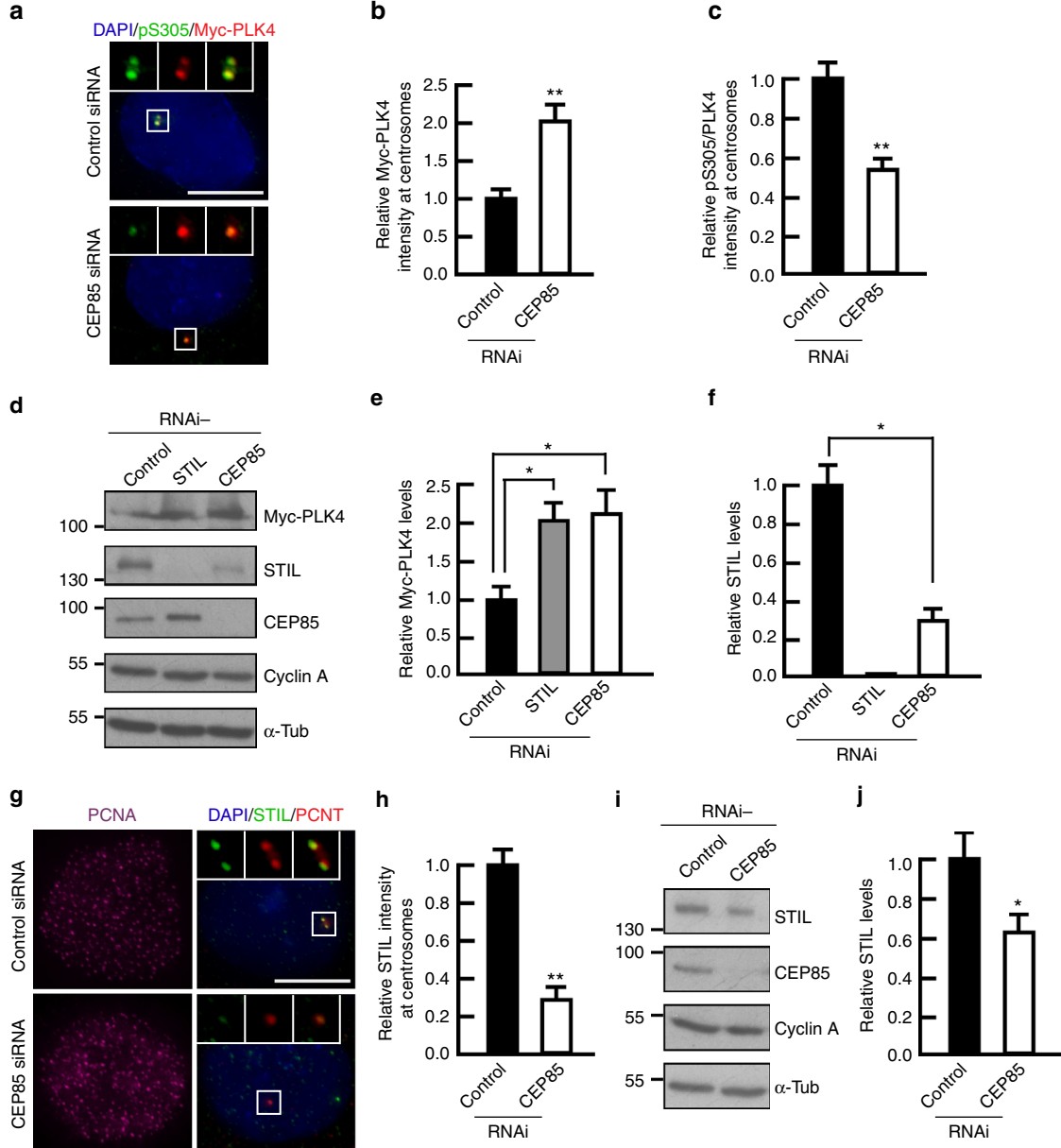

**Fig. 2** CEP85 is required for PLK4 activation and STIL localization to centrosomes. **a–c** Examining the impact of CEP85 depletion on PLK4 activation using the PLK4-induced centriole overduplication assays (see Methods). Selected images showing Myc-PLK4 and PLK4 pS305 labelling. Scale bar 10 μm, white boxes indicate the magnified region. **b**, **c** The graph indicates the levels of Myc-PLK4 and the relative ratio of pS305/PLK4 at centrosomes after depletion of endogenous CEP85. (n = 200/experiment, three independent experiments). **d–f** Western blot showing the levels of Myc-PLK4, CEP85 and STIL in control or CEP85 siRNA transfected cells. Cyclin A was used as a cell cycle marker and α-tubulin served as a loading control. **e**, **f** Quantification of protein levels shown in D (α-tubulin normalized, n = 2/experiment, five independent experiments,). **g**, **h** IF analysis of STIL localization in control or CEP85-depleted cells. The S-phase arrest assays were performed as described in Methods, followed by labelling with DAPI and the indicated antibodies. Scale bar 10 μm, white boxes indicate the magnified region. **h** Quantification showing the relative levels of STIL at centrosomes in PCNA-positive cells (n = 200/experiment, three independent experiments). **i**, **j** Western blot indicates the levels of STIL and CEP85 in control or CEP85 siRNA treated cells. Cyclin A and α-tubulin served as a cell cycle marker and loading control, respectively. **j** Quantification of the indicated protein levels shown in I (α-tubulin normalized, n = 2/experiment, five independent experiments). Two-tailed t-test was performed for all p-values, all error bars represent SD, and asterisks for p-values are **p < 0.01 and *p < 0.05

resistant CEP85 (Fig. 1a–c). Indicative of decreased centriole numbers, this was accompanied by a decrease in the number of Centrin foci in G2 cells, a phenotype we could rescue by expressing an RNAi-resistant CEP85 (Fig. 1d–f). Pronounced centriole duplication defects were also observed using PLK4-induced centriole overduplication assays (Fig. 1g, h), S-phase arrest induced centriole overduplication assays (Fig. 1i, j), and upon transient CRISPR-mediated disruption of the CEP85

genomic locus to levels akin to CEP192 KO cells (Supplementary Fig. 1e-g). Together, these multiple lines of evidence indicate that CEP85 is a positive regulator of centriole duplication.

Next, we sought to determine at which step CEP85 operates during daughter centriole assembly. To address this question, we depleted endogenous CEP85 in U-2 OS cells expressing Tet-inducible PLK4 which allows for the assembly of multiple daughter centrioles in the vicinity of mother centrioles[33].

Post-transfection, cells were enriched in S-phase through the addition of hydroxyurea, PLK4 expression was induced and cells were fixed and submitted to immunofluorescence (IF) analysis to assay for the recruitment of CEP192, PLK4, STIL, SASS6 and Centrin. We found that while CEP85 depletion did not significantly affect CEP192 levels, increased centrosomal levels of PLK4 were observed along with a robust decrease of STIL and the downstream factors SASS6 and Centrin in the vicinity of mother centrioles compared to the control (Fig. 1k, l). To define the requirements for CEP85 localization to the centrioles, we depleted endogenous CEP192, CEP152, PLK4 and STIL in U-2 OS cells using the same assay. We found that depletion of CEP192, CEP152 and PLK4 all led to a reduction in centriolar recruitment of CEP85, while STIL depletion led to a ~20% increase in CEP85 levels at centrioles (Supplementary Fig. 2f, g), indicative of potential feedback regulation of CEP85 levels that warrants further investigation. Together, these data suggest that CEP85 acts downstream of CEP192, CEP152 and PLK4 and plays an important role in the regulation of PLK4 and STIL recruitment.

**CEP85 is required for STIL localization and PLK4 activation.**
PLK4 activation at centrioles is a key step in centriole duplication.

Indeed, upon autophosphorylation at its activation loop, active PLK4 phosphorylates itself on an evolutionary conserved phosphodegron resulting in β-TrCP-mediated PLK4 degradation[36]. Since, we observed increased PLK4 abundance in the absence of CEP85, we hypothesized that CEP85 depletion could negatively impact PLK4 autophosphorylation thereby stabilizing it. To test this hypothesis, we depleted CEP85 in synchronized cells and assayed PLK4 activity by quantitative IF analysis using a phospho-specific PLK4 antibody that recognizes its autophosphorylated residue S305 as a proxy for PLK4 activity[51]. Inhibition of PLK4 autophosphorylation using the potent PLK4 inhibitor centrinone, demonstrated the specificity of the antibody in this assay (Supplementary Fig. 2a, b)[52]. While, we observed that upon CEP85 depletion centrosomal levels of PLK4 were increased twofold, the relative levels of active PLK4 (pS305/PLK4) were reduced (Fig. 2a–c). This inactive pool of PLK4 appears to be stabilized as we consistently observed a significant increase in the total cellular levels of PLK4 upon CEP85 depletion (Fig. 2d, e). We had observed that STIL levels are reduced upon CEP85 depletion using the PLK4-induced centriole overduplication assay (Figs. 1k, l, 2f). To determine whether the global levels of STIL are also affected by lowered steady-state levels of CEP85 in the absence of PLK4 overexpression, we depleted CEP85 in U-2 OS cells using the same siRNA. Transfected cells were exposed to

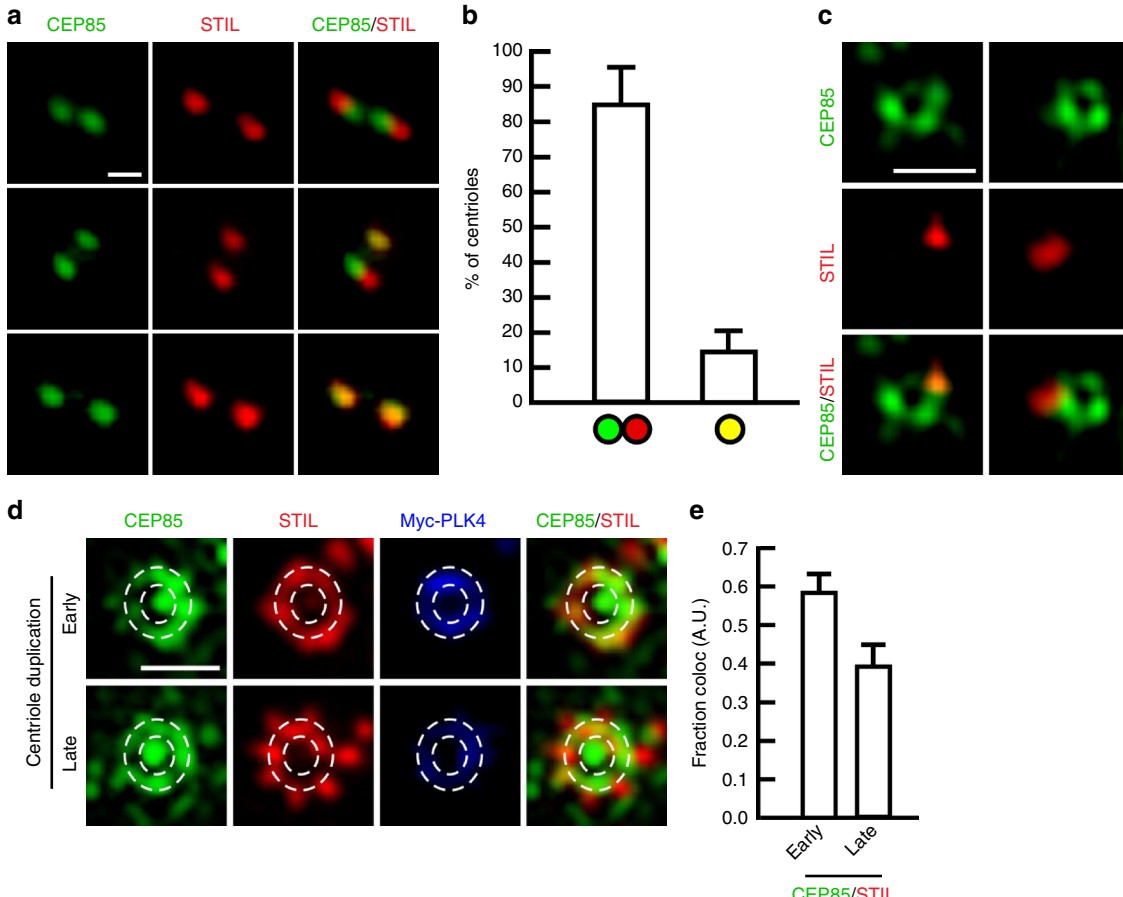

**Fig. 3** CEP85 co-localizes with STIL at an early stage of centriole duplication. **a, b** IF analysis of CEP85 and STIL localization in U-2 OS cells after 24 h S-phase arrest using thymidine (1 mM). Selected images showing CEP85 and STIL labelling. **b** Bar graph, the percentage of centrioles with different localization patterns (n = 100/experiment, three independent experiments). Scale bar 0.5 μm. **c** 3D-SIM micrographs of S-phase arrested (1 mM thymidine) U-2 OS cells stained with the indicated antibodies. **d, e** 3D-SIM micrographs in U-2 OS Tet-inducible Myc-PLK4 cells after 24 h PLK4 induction and S-phase arrest using thymidine (1 mM), showing CEP85, STIL and Myc-PLK4 staining. PLK4 ring structures were encircled by white dashes. Scale bar 0.5 μm. **e** The graph indicates the fraction co-localization between CEP85 and STIL (n = 20/experiment, two independent experiments). All error bars represent SD

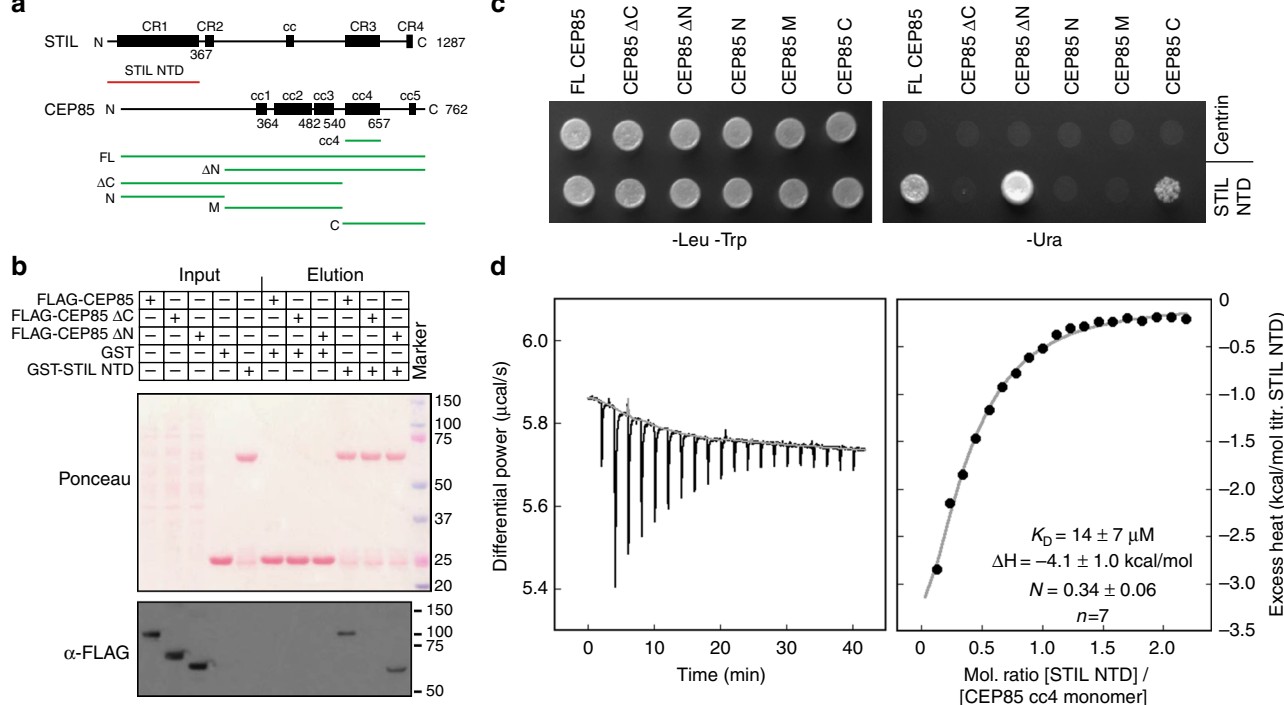

**Fig. 4** The N-terminal domain of STIL directly interacts with a C-terminal coiled-coil domain of CEP85. **a** Domain overview of human STIL and CEP85. CR conserved region; cc coiled-coil; NTD N-terminal domain. Constructs that were used in this work are indicated by red and green lines. **b** STIL NTD and CEP85 interact. The C-terminal region of CEP85 is required for this interaction. Western blot showing a pull-down experiment with immobilized, recombinant GST or GST-human STIL NTD and lysates from tissue culture cells overexpressing the indicated 3xFLAG-tagged CEP85 constructs. **c** The C-terminal region of CEP85 shows a yeast-two-hybrid interaction with human STIL NTD. Yeast transformed with the indicated bait and prey constructs were plated on SC-Leu/-Trp plates (selecting for bait and prey plasmid, left) and on SC-Ura plates (selecting for Ura promoter activation, right). **d** Recombinant chicken STIL NTD and human CEP85 cc4 directly interact with micromolar affinity. Typical ITC of chicken STIL NTD and human CEP85 cc4 binding at 25 ° C. The resulting $K_D$, $\Delta H$ and STIL NTD/CEP85 cc4 binding stoichiometry ($N$) as an average from a total of seven independent measurements are indicated (±standard deviation)

hydroxyurea to enrich them in S-phase prior to fixation and immunolabelling for STIL and PCNT to mark the position of centrioles. Consistent with the PLK4 overexpression data (Fig. 1k, l), we found a marked decrease of centrosomal level of STIL upon CEP85 depletion (Fig. 2g, h). Defects in STIL recruitment coincided with a slight decrease in the total cellular levels of STIL (Fig. 2i, j), indicating that CEP85 depletion might also lead to a decrease in the total amount of bioavailable STIL. Previous results have indicated that STIL is an upstream regulator of PLK4 and that it contributes to its full activation[23]. Taken together with our results, these data suggest that CEP85 may play a direct role in the regulation of STIL to facilitate full activation of PLK4 kinase and we sought to examine this possibility.

**CEP85 and STIL recruitment during centriole duplication.** Analysis of the spatial distribution of CEP85 during centriole duplication indicated that it localized in the vicinity of CEP192, at the proximal end of mother centrioles in interphase cells (Supplementary Fig. 2e). STIL is known to be degraded in G1 and recruited again to the site of procentriole assembly during S-phase, at the onset of centriole duplication[41] and for this reason we examined the spatial distribution of STIL and CEP85 during centriole duplication in S-phase. As cells progress into S-phase, we observed ~15% of centrioles with an overlapping CEP85 and STIL localization and 85% of centrioles where CEP85 and STIL were found adjacent to but optically resolvable from each other (Fig. 3a, b). To further examine the spatial distribution of CEP85 and STIL beyond the diffraction limit we used 3D-SIM imaging

and found that CEP85 accumulated around mother centrioles, with a small fraction of cells harbouring a dot-like STIL pattern that overlapped with CEP85 (Fig. 3c), suggesting that CEP85 may only transiently associate with STIL at centrioles. To test this hypothesis, we induced centriole overduplication by overexpressing PLK4 in S-phase arrested cells and used IF analyses to measure the spatial distribution of STIL, PLK4 and CEP85 at early and later stages of centriole formation[53]. Analysis of 3D-SIM data sets indicated that the proportion of CEP85 that overlaps with STIL is the highest early on during procentriole assembly, prior to procentriole assembly and elongation (Fig. 3d, e). Taken together, these results support the notion that CEP85 associates with STIL to regulate robust activation of PLK4 early during centriole duplication.

**CEP85 interacts directly with the N-terminal domain of STIL.** Previous work on the CPAP-STIL complex raised the possibility that the N-terminal domain (NTD) of STIL is located towards the periphery of the centriole cylinder[54, 55] and therefore might be found close to CEP85. To test whether both could interact with each other, we first used a pull-down experiment with immobilised recombinant GST-STIL NTD and cell lysates expressing different FLAG-tagged CEP85 constructs (Fig. 4a, b). In this assay, we found that the STIL NTD indeed was able to associate with CEP85 and that binding depended on the C-terminal region of CEP85. Yeast-two-hybrid assays suggested that this region binds directly to the STIL NTD (Fig. 4a, c). Additional isothermal titration calorimetry (ITC) (Fig. 4d, Supplementary Fig. 5g) and

**Table 1 Data collection and refinement statistics**

| | *H. sapiens* CEP85[570-656] | SeMet *T. adhaerens* STIL[1-348] | Complex: SeMet *T. adhaerens* STIL[1-348] - *H. sapiens* CEP85[570-656] |
|---|---|---|---|
| **Data collection** | | | |
| Space group | *I*222 | *P*2₁ | *I*4₁32 |
| *Cell dimensions* | | | |
| *a, b, c* (Å) | 37.9, 73.9, 128.1 | 66.0, 75.2, 68.5 | 268.7, 268.7, 268.7 |
| *α, β, γ* (°) | 90.0, 90.0, 90.0 | 90.0, 97.2, 90.0 | 90.0, 90.0, 90.0 |
| Resolution (Å) | 36.98-1.67 (1.76-1.67)[a] | 37.62-2.09 (2.14-2.09) | 95.00-4.60 (5.14-4.60) |
| $R_{merge}$ | 0.202 (1.552) | 0.160 (0.861) | 0.272 (3.033) |
| $R_{pim}$ | 0.063 (0.506) | 0.077 (0.416) | 0.043 (0.470) |
| *I*/σ*I* | 8.6 (1.4) | 7.8 (2.0) | 13.2 (1.9) |
| Completeness (%) | 98.9 (92.2) | 100.0 (100.0) | 100.0 (100.0) |
| Redundancy | 11.1 (10.1) | 5.2 (5.2) | 39.9 (42.1) |
| **Refinement** | | | |
| Resolution (Å) | 36.98-1.67 | 37.62-2.09 | 95.00-4.60 |
| No. of reflections | 21,329 | 39,724 | 9507 |
| $R_{work}$/$R_{free}$ | 20.3/23.9 | 19.7/24.1 | 35.3/36.5 |
| *No. of atoms* | | | |
| Protein | 1471 | 5247 | 1829 (poly-ala model) |
| Ligand/ion | 22 (MPD) | 12 (MES) | 0 |
| Water | 227 | 294 | 0 |
| *B-factors* | | | |
| Protein | 22.7 | 28.5 | 271.8 |
| Ligand/ion | 28.5 (MPD) | 36.4 (MES) | NA |
| Water | 33.9 | 28.5 | NA |
| *R.m.s. deviations* | | | |
| Bond length (Å) | 0.002 | 0.003 | 0.003 |
| Bond angles (°) | 0.538 | 0.573 | 0.896 |
| **Validation** | | | |
| Fo,Fc correlation | 0.96 | 0.94 | 0.86 |
| Molprobity score | 1.1 (99th percentile) | 1.2 (100th percentile) | 1.3 (100th percentile) |
| Molprobity clashscore | 3.0 | 2.4 | 2.7 |
| Poor rotamers (%) | 0.0 | 0.2 | NA (poly-ala model) |
| Ramachandran outliers (%) | 0.0 | 0.0 | 0.3 |
| Ramachandran favoured (%) | 100.0 | 96.6 | 95.9 |
| PDB accession code | 5OI7 | 5OI9 | 5OID |

Data sets were obtained from single crystals
[a] Values in parenthesis are for highest resolution shell

analytical ultracentrifugation (AUC) experiments (Supplementary Fig. 4c) with recombinant proteins allowed us to refine the binding region of CEP85 to a coiled-coil domain within this C-terminal region (cc4) and, depending on the experimental conditions, suggested a binding affinity of ~14–65 μM for this construct (Fig. 4d, Supplementary Fig. 4c, Supplementary Fig. 5g). These ITC and AUC experiments were performed with chicken STIL NTD (66% identical to human STIL NTD, Supplementary Fig. 8b) since we found human STIL NTD to have an aggregation tendency at room temperature. NMR (Supplementary Fig. 3d) and native, as well as cross-linking mass-spectrometry data (Supplementary Fig. 4d, Supplementary Fig. 6) support the notion of a direct interaction between the STIL NTD and the CEP85 cc4 domain. Further biophysical characterisations of the individual proteins in solution suggest that the STIL NTD is monomeric (Supplementary Fig. 4c, Supplementary Fig. 5a), while the CEP85 cc4 is in a monomer-dimer equilibrium at room temperature and at the concentrations and buffer conditions used in these biophysical studies (Supplementary Fig. 5b) while becoming predominantly dimeric at reduced temperatures (Supplementary Fig. 4c, Supplementary Fig. 5f).

**Structural characterisation of the CEP85-STIL interaction region.** To further characterise the CEP85-STIL interaction, we sought to obtain higher resolution information of this complex.

First, we determined the structures of the STIL NTD (from *Trichoplax adhaerens*, 28% identical to *Homo sapiens* STIL NTD, Supplementary Fig. 8b) and *Homo sapiens* CEP85 cc4 (human) at a resolution of 2.1 and 1.7 Å, respectively using X-ray crystallography (Table 1, Supplementary Table 1, Fig. 5a, b).

The NTD of STIL revealed a global structural similarity to the N-terminal domain of the autophagy-related protein ATG7 (Supplementary Fig. 3a), a protein–protein interaction domain that forms binary complexes with Atg10 and Atg3[56]. It is composed of two subdomains that are structurally akin to each other suggesting that this domain might have evolved via a domain duplication and fusion event (Supplementary Fig. 3b). The common structural core of both subdomains consists of a six-stranded open β-barrel capped at the top and on one side with α-helices, a fold that resembles the so-called MPN/JAB1/PAD-1/JAMM domain, as observed, for example, in eIF3f (Supplementary Fig. 3b). The STIL NTD subdomain 1 is structurally more akin to MPN domains than the subdomain 2 and this similarity extends to the position and length of the capping helix and the presence of a crossover hairpin-loop, connecting strands 4 and 7. Interestingly, we found that these capping helices and their vicinity in both STIL NTD subdomains define the most conserved patches on their surface (Fig. 5a) making them prime candidates for a role in protein–protein interactions, in this instance binding to CEP85 cc4.

The structure of CEP85 cc4 revealed a parallel coiled-coil dimer (Fig. 5b, multiple sequence alignment in Supplementary Fig. 8c) that is stabilized through the packing of hydrophobic residues in a canonical knobs-into-holes fashion. Additional sidechain–sidechain hydrogen bonds and salt bridges act to further stabilize the dimer interface. As with the STIL NTD, we calculated the sequence conservation of the CEP85 coiled-coil (cc4) and mapped it onto its structure. This analysis showed the presence of a striking conservation on the outside of the coiled-coil towards its C-terminal end (Fig. 5b, within the dashed box)

suggesting that this region might be involved in mediating protein–protein interactions.

Subsequently, we also determined the structure of the complex between *Trichoplax adhaerens* STIL-NTD and human CEP85 cc4 by X-ray crystallography to a resolution of 4.6 Å (Table 1, Supplementary Tables 2, 3, Fig. 5c). We did not obtain diffraction-quality crystals with STIL-NTD homologues from other species that we tested. The structure of the complex was solved by MAD using crystals of the corresponding seleno-methionine derivative (Supplementary Fig. 4a). At this resolution,

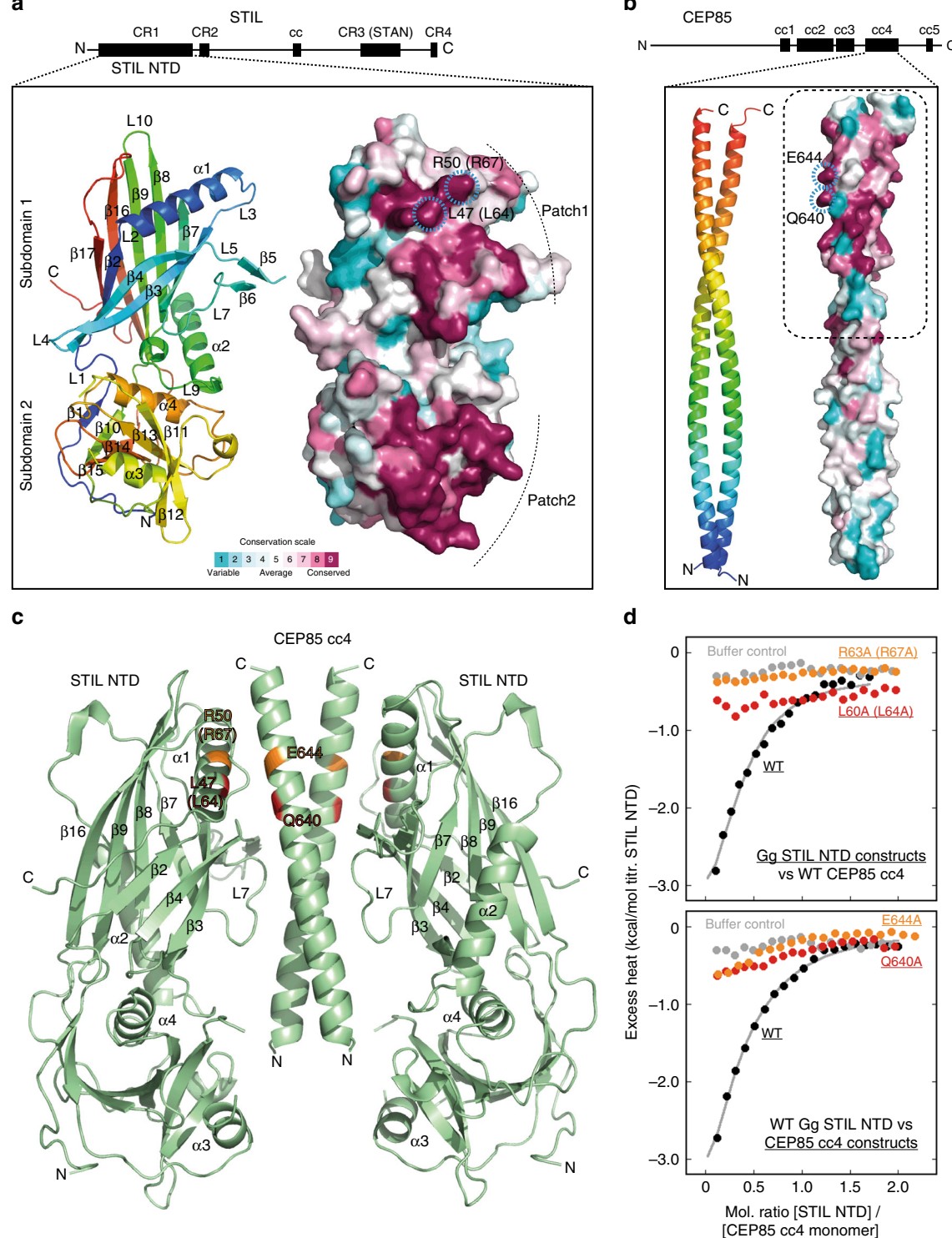

side-chains cannot be discerned. The part of the CEP85 coiled-coil that is visible in the electron density map lacked methionines that could serve as landmarks to obtain its correct register and orientation in the structure. To obtain these landmarks, we introduced methionines into its sequence, crystallized the corresponding selenomethionine derivates of their complexes with *Trichoplax adhaerens* STIL-NTD (Supplementary Table 3), and subsequently calculated phased anomalous difference maps (Supplementary Fig. 4b) from the corresponding data sets to locate the positions of the introduced selenomethionines.

The structure of the CEP85-STIL complex shows a 2:2 binding ratio with one STIL NTD bound to each of the two coiled-coil strands of the parallel CEP85 cc4 dimer (Fig. 5c, Supplementary Fig. 4b). Compared with the unbound CEP85 coiled-coil structure (Fig. 5b) we did not observe any density for the N-terminal half of the coiled-coil, probably due to its partial unfolding in the crystal. In solution, at reduced temperatures that stabilize the CEP85 cc4 dimer (Supplementary Fig. 5b, Supplementary Fig. 5f), both ITC binding studies and analytical ultracentrifugation with human CEP85 cc4 and chicken STIL NTD are in agreement with a 2:2 binding stoichiometry between both proteins (Supplementary Fig. 4c, Supplementary Fig. 5g). Furthermore, cross-linking mass-spectrometry experiments with the corresponding 2:2 complex revealed the presence of a specific cross-link close to the binding interface observed in our crystal structure (Supplementary Fig. 6). Finally, native mass-spectrometry experiments with both proteins at room temperature (Supplementary Fig. 4d) confirm that the CEP85 cc4 needs to be dimeric to engage the STIL NTD. Together, these data suggest that the binding mode between STIL NTD and CEP85 cc4 is conserved across species.

The interaction between the CEP85 coiled-coil and STIL NTD involves mainly subdomain 1 of STIL, in particular the region of the β-barrel capping helix and the crossover hairpin-loop that, together, constitute one of the two conserved patches in the STIL NTD (Fig. 5a, c, Supplementary Fig. 3c). Each of the STIL NTDs interacts with both CEP85 coiled-coil helices. The crossover hairpin-loop of subdomain 1 of STIL NTD contacts towards the N-terminal part of one of the coiled-coil helices, whereas its barrel capping helix contacts towards the C-terminal end of the other CEP85 coiled-coil helix. Due to the relatively low resolution of the complex structure we could not resolve any sidechains. However, several conserved residues of the CEP85 coiled-coil and STIL subdomain 1 would be well placed to make electrostatic and hydrophobic interactions with each other (Fig. 5a–c, Supplementary Fig. 8b-d).

Docking of the high-resolution CEP85 cc4 structure into the electron density map of the complex suggest that the coiled-coil region of CEP85 cc4 that is invisible in the complex could potentially interact with the second conserved patch of the STIL NTD (constituted by the capping helix and the crossover loop-helix from its subdomain 2) (Supplementary Fig. 3c). However, NMR experiments with the chicken STIL NTD R63A (R67A in human STIL) point mutation that compromises the interaction between subdomain 1 of STIL NTD and human CEP85 cc4 (Fig. 5d) did not provide evidence for a remaining (potentially weak) interaction (Supplementary Fig. 3d). Thus, binding of STIL NTD to CEP85 cc4 occurs probably exclusively through subdomain 1 of STIL NTD.

**CEP85-STIL interaction is required for STIL localization.** The structure of the complex between STIL NTD and CEP85 cc4 indicated that the highly conserved residues L47 and R50 in *Trichoplax adhaerens* STIL (L64 and R67 in human STIL, L60 and R63 in *Gallus gallus* STIL) and human CEP85 Q640 and E644 are located at their interaction interface (Fig. 5a–c). When we introduced single-alanine mutations into the recombinant proteins at these positions, we found that binding was strongly reduced in vitro as judged by ITC (Fig. 5d) and NMR experiments (Supplementary Fig. 3d). This was due to a direct effect on the binding interface as the structure of the proteins was retained as indicated by CD (Supplementary Fig. 5d-f) and NMR spectra (Supplementary Fig. 3d). In pull-down assays with cell lysates, binding of recombinant GST-STIL NTD to full length CEP85 was also strongly compromised in the corresponding mutants (Fig. 6a, Supplementary Fig. 7a).

To further confirm that these observations are also relevant for the interaction of the full-length proteins in vivo, we artificially targeted mCherry-tagged CEP85 wild type, Q640A or E644A to microtubules by fusing them in frame to the microtubule-binding domain of MAP7 (3xFLAG-MAP7[17–282]-mCherry-CEP85. MAP7[17–282] comprises MAP7's microtubule binding domain)[57] and then asked whether 3xStrep-GFP-tagged STIL would follow CEP85 to the microtubule cytoskeleton. A comparable system based on the microtubule binding protein Tau was used previously to probe protein–protein interactions in vivo[58]. The results shown in Fig. 6h, i demonstrate that wild-type CEP85 robustly recruited STIL to microtubules. In stark contrast, the CEP85 Q640A, E644A and double mutants deficient in STIL binding were strongly compromised in their ability to recruit STIL when expressed at WT-like levels (Fig. 6h–j).

Next, we set out to determine whether these two key residues of CEP85 were also functionally important for the role of CEP85 in centriole duplication. To this end, we first depleted endogenous CEP85 in U-2 OS cells and tested the ability of RNAi-resistant wild-type CEP85 and CEP85 Q640A, CEP85 E644A or CEP85 Q640A and E644A double mutations to rescue the centriole duplication defect. We found that this panel of CEP85 mutants did not appear to adversely affect the overall levels of CEP85 or its

**Fig. 5** STIL NTD interacts with CEP85 cc4 through a conserved interface. **a** High-resolution structure of *Trichoplax adhaerens* STIL NTD. Left, as ribbon presentation, rainbow coloured from N- to C-terminus. α-helices (α), β-sheets (β) and loops (L) are labelled consecutively. Right, equivalent view as molecular surface coloured by CONSURF evolutionary conservation score (right) from unconserved (cyan) to highly conserved (burgundy), revealing the presence of two conserved patches on the surface of *Trichoplax adhaerens* STIL NTD. Ringed in blue are residues L47 and R50 (L64 and R67 in human STIL) that are part of patch 1 and are located at the interaction interface in the CEP85-STIL complex as shown in **c**. **b** High-resolution structure of human CEP85 cc4 as ribbon presentation (rainbow coloured from N- to C-terminus) (left) and as molecular surface coloured by CONSURF evolutionary conservation score (right). The dashed box indicates the CEP85 cc4 region with visible electron density in the CEP85-STIL complex (shown in **c**). Residues Q640 and E644 that are found at the interface of this complex are ringed in blue. **c** Complex of *Trichoplax adhaerens* STIL NTD and human CEP85 cc4 at a resolution of 4.6 Å, as a ribbon presentation. The correct register of the CEP85 cc4 was obtained by introducing methionines into CEP85 cc4 and calculating phased anomalous difference maps from corresponding selenomethionine-derivate data sets (Supplementary Fig. 4b). The human equivalents of the *Trichoplax adhaerens* STIL residues are indicated in brackets. **d** Mutation of conserved interface residues in STIL NTD and CEP85 cc4 compromises binding affinity. ITC measurements of chicken STIL NTD titrated into human CEP85 cc4 at 25 °C. Excess heat observed for the mutant proteins approach that obtained when titrating wild-type (WT) STIL NTD into buffer alone as a non-binding control. Conserved residues mutated in chicken STIL NTD are L60A and R63A, equivalent to L64 and R67A in human STIL (in brackets) and L47 and R50 in *Trichoplax adhaerens* (Fig. 5a)

ability to localize at centrosomes (Fig. 6b–d). Subsequently, we assayed for the effect of these mutants on STIL recruitment to centrioles and for the ability of centrioles to duplicate. Our results demonstrated that in the absence of a functional CEP85-STIL

binding interface both centriole duplication and STIL recruitment to centrosomes were perturbed (Fig. 6e–g).

Conversely, we then asked whether defects in STIL recruitment and centriole duplication would also be observed with the STIL

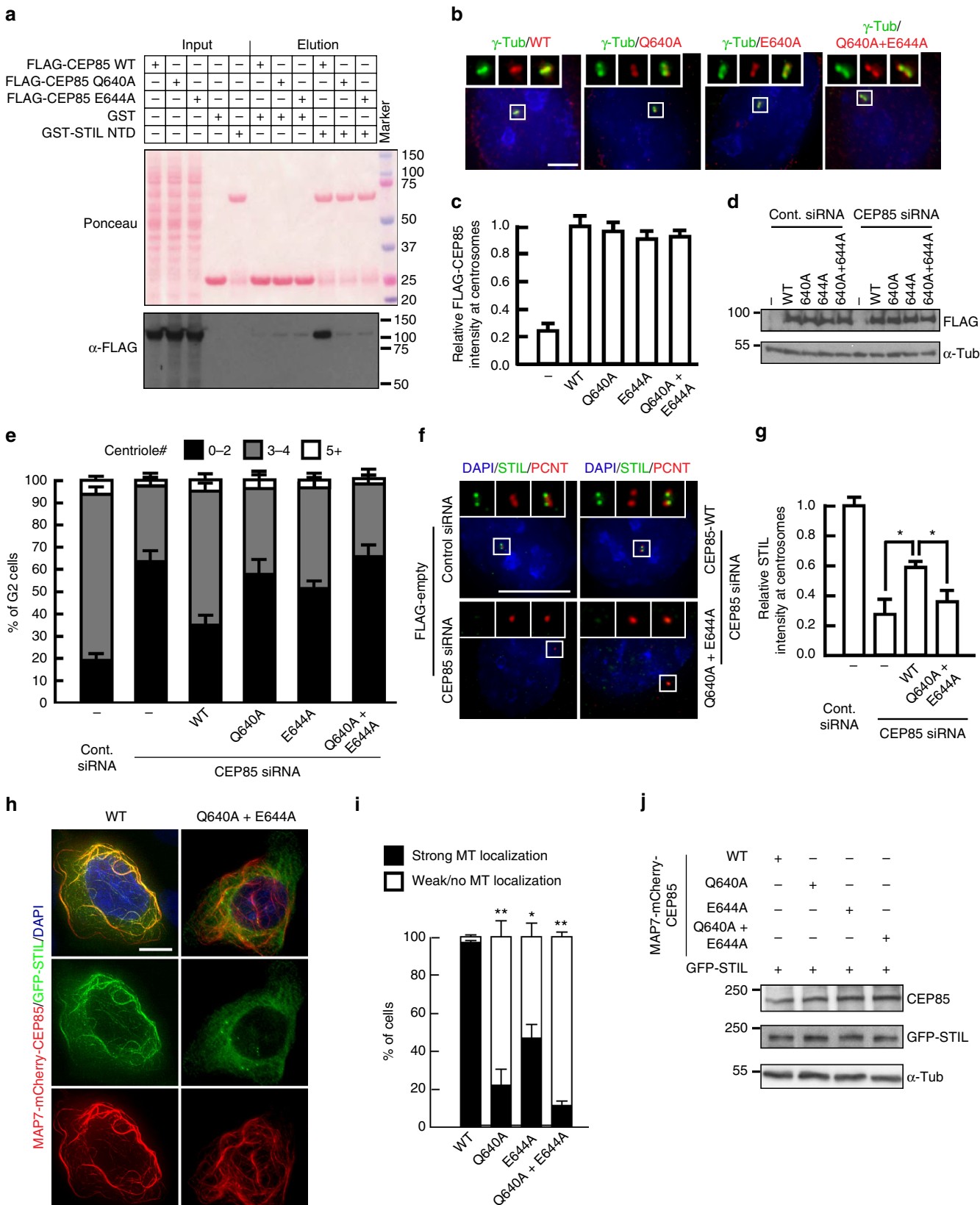

point mutations that are defective for CEP85 binding. First, we used the L64A and R67A double mutant of STIL and showed that, compared to wild-type STIL, the mutant form displays significantly reduced centriole localization, further substantiating a role of the described CEP85-STIL interaction in facilitating STIL recruitment to centrioles (Supplementary Fig. 7b, c). We then assayed for the ability of the L64A, R67A, as well as the L64A and R67A STIL mutants to act in centriole duplication. To this end, we established a STIL overexpression system in STIL CRISPR knockout cells (that do not have centrioles) and assayed for de novo centriole formation in these cells. Our results indicated that both STIL L64A and R67A mutants were not able to restore centriole biogenesis to levels comparable to wild-type STIL (Supplementary Fig. 7d, e). We also investigated the ability of these STIL mutants to rescue centriole duplication at lower (inducible) STIL expression levels in cells that were siRNA depleted for STIL. Consistently, compared to wild-type STIL, the expression of these STIL mutants could not fully rescue the centriole duplication defect in STIL depleted cells (Supplementary Fig. 7f).

We had observed that depletion of CEP85 leads to a decrease in total cellular levels of STIL raising the possibility that the requirement for CEP85 could be bypassed by STIL overexpression. To test this possibility, we depleted endogenous CEP85 in U-2 OS cells and tested the ability of WT STIL and the STIL L64A and R67A mutant to rescue centriole duplication. Our results indicate that expression of WT STIL and the STIL mutant were unable to rescue centriole duplication (Supplementary Fig. 9a-c). Consistent with this observation, we found that the overexpression of a non-degradable form of the STIL L64A and R67A mutants (that are compromised in their ability to bind CEP85), containing the previously described Val1219X mutation[43], was unable to induce centriole overduplication to WT STIL levels (Supplementary Fig. 9d-f) thereby supporting a dual role for CEP85 in the recruitment of STIL to the site of centriole assembly and its stability.

**CEP85 and STIL binding is required for PLK4 activation**. Taking into account the recently described role of STIL in modulating PLK4 activity[23], we asked whether perturbing the CEP85-STIL binding interface would modulate PLK4 activity in vivo. To test this hypothesis, we used lentiviruses to stably express eGFP-tagged wild-type CEP85 and the CEP85 Q640A and E644A double mutant in PLK4 overexpressing cells and measured the impact on centriole overduplication. Our results showed that under these conditions, over 70% of cells overexpressing GFP alone were able to overduplicate their centrioles.

Although the expression of wild-type CEP85 had a minor negative effect on centriole duplication, expression of comparable levels of the CEP85 Q640A and E644A double mutant had a more pronounced suppressive effect on PLK4's ability to drive centriole amplification, which was accompanied by an increase in the total cellular levels of PLK4 (Fig. 7a–d). Next, we synchronized cells in S-phase and assessed PLK4 activity by immunofluorescence using the autophosphorylated S305 residue on PLK4 as a proxy for kinase activity[51]. Consistent with our previous results of CEP85 depletion (Fig. 2a–c), we found that expression of the CEP85 Q640A and E644A double mutant also increased the centrosomal levels of PLK4, and at the same time, led to decreased PLK4 activity levels as judged by a lowered ratio of pS305/PLK4 levels (Fig. 7e–g). Thus, our results suggest that the CEP85-STIL interaction is required to enable full PLK4 kinase activation in vivo to drive centriole duplication.

## Discussion

A finely regulated centriole duplication process is essential for faithful cell division. Dysfunctions in this process can result in numerous human diseases, such as microcephaly and cancer. Despite progress into elucidating the molecular mechanisms of centriole assembly, many aspects of this process are still not fully understood. Here we demonstrate that CEP85 is essential for faithful centriole duplication through a direct interaction with the previously uncharacterised N-terminal domain of STIL, a key factor involved in the early steps of centriole formation. The determination of the CEP85:STIL complex structure and a subsequent structure-guided functional characterisation of this complex in vivo allowed us to uncover a previously undescribed mode of regulation of the early steps in centriole assembly.

Several lines of evidence suggest that the CEP85-STIL interaction is physiologically relevant: (1) Biochemically, binding between both proteins is observed across different species; (2) The in vitro binding affinities of CEP85/STIL are within the range of physiologically relevant physical interactions; (3) Structurally, the interaction interface contains conserved residues on both sides; (4) Conserved interface residues on both proteins, when mutated, compromise the CEP85-STIL interaction and result in centriole duplication defects; (5) CEP85 and STIL show a partial co-localization at the site of daughter centriole assembly and robust targeting of STIL to centrosomes requires a functional CEP85-STIL interface.

The canonical view of STIL's role in centriole duplication places it at an early stage of centriole formation, when PLK4 has been recruited and subsequently restricted to a single dot on one side of the mother centriole. The binding of STIL then activates

**Fig. 6** The interaction between CEP85 and STIL is essential for centriole duplication. **a** The interface residues described in Fig. 5 are required for an efficient interaction of human STIL NTD and CEP85. Western blot showing a pull-down experiment with immobilized, recombinant GST or GST-human STIL NTD, and lysates from cells overexpressing 3xFLAG-tagged CEP85 (WT and mutants) as indicated. **b**, **c** IF analysis of CEP85 localization. Tetracycline and hydroxyurea were added to induce the expression of FLAG-CEP85 transgenes and to arrest cells in S-phase for 24 h before fixation. Cells were labelled with the indicated antibodies. Scale bar 10 μm, white boxes indicate the magnified region. **c** Quantification showing the relative levels of FLAG-CEP85 (WT and mutants) at centrosomes. **d** Western blot analysis of the indicated protein levels. α-tubulin served as a loading control. **e** Effect of CEP85 mutations on centriole duplication. U-2 OS cells conditionally expressing FLAG or the siRNA-resistant FLAG-CEP85 (WT and mutants) were treated with control or CEP85 siRNA and induced with tetracycline for 72 h. The G2-phase arrest assays were performed as described in Methods. Quantification showing the percentage of cells with the indicated centriole number ($n = 100$/experiment, three independent experiments). **f**, **g** The role of CEP85 mutations in STIL localization. The S-phase arrest assays (see the Methods) were performed using U-2 OS cells expressing Tet-inducible FLAG or the siRNA-resistant FLAG-CEP85 (WT and mutants) and tetracycline was added for 72 h. Scale bar 10 μm, white boxes indicate the magnified region. **g** Quantification showing the relative levels of STIL at centrosomes ($n = 100$/experiment, three independent experiments). **h**, **i** Confirming the interaction between CEP85 and STIL in vivo. U-2 OS STIL CRISPR KO cells were co-transfected with GFP-STIL and MAP7-mCherry-CEP85 (WT and mutants) for 24 h. Cells were fixed with 4% PFA and stained with DAPI. **i** The graph indicates the percentage of cells with STIL recruited to microtubules. **j** Western blot shows the levels of GFP-STIL and MAP7-mCherry-CEP85 (WT and mutants). Two-tailed t-test was performed for all p-values, all error bars represent SD, and asterisks for p-values are **$p < 0.01$ and *$p < 0.05$

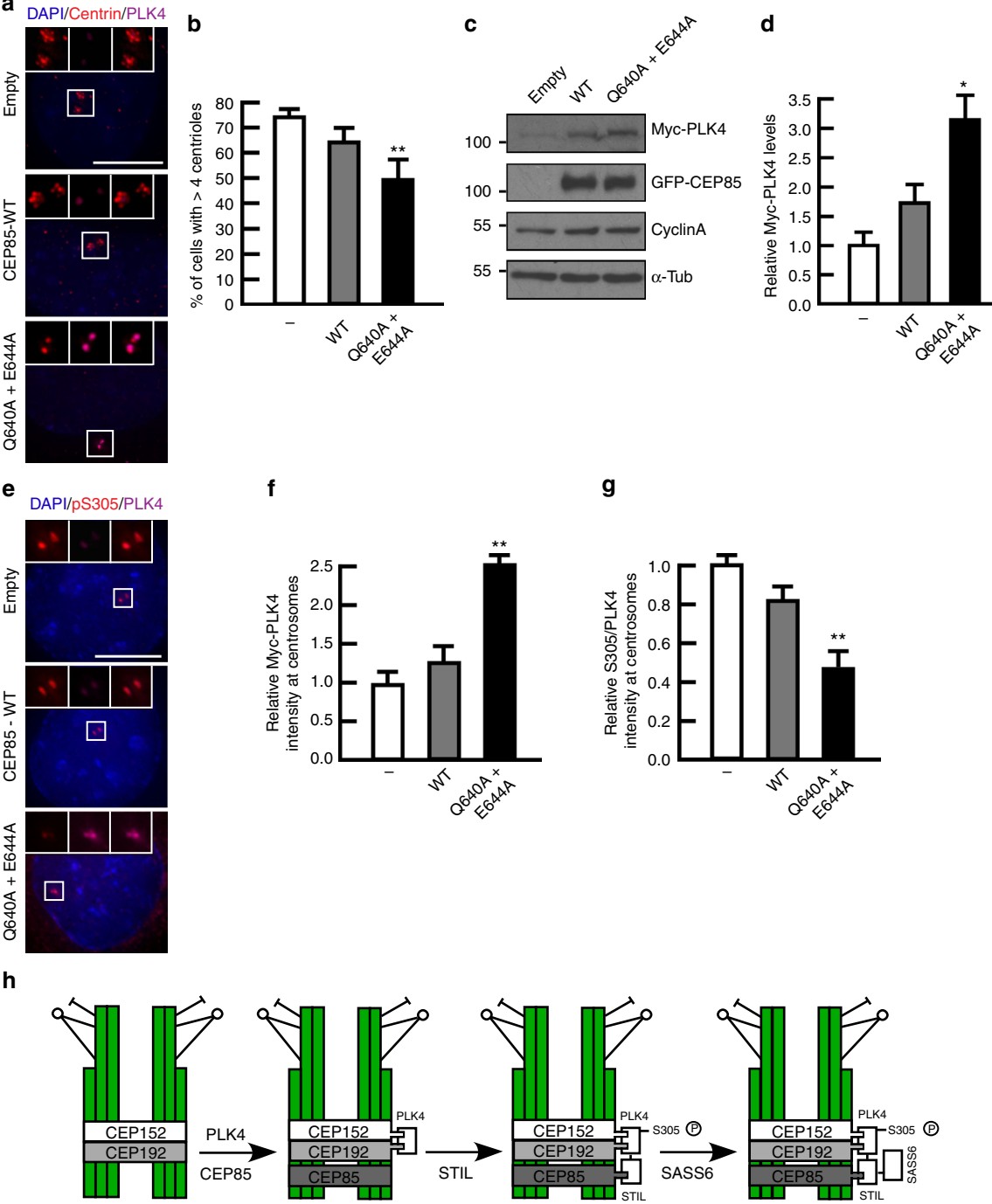

**Fig. 7** The interaction between CEP85 and STIL is essential for PLK4 activation and PLK4-induced centriole overduplication. **a, b** Effect of CEP85 mutations on centriole overduplication. U-2 OS cells conditionally expressing Myc-PLK4 and constitutively expressing GFP-CEP85 (WT and mutant) were used to perform the PLK4-induced centriole overduplication assays. Cells were labelled with DAPI and the indicated antibodies. Scale bar 10 μm, white boxes indicate the magnified region. **b** Quantification showing the percentage of cells with over four centrioles ($n = 100$/experiment, three independent experiments). **c** Western blot analysis indicating the Myc-PLK4 and GFP-CEP85 protein levels using the PLK4 assays described in A. Cyclin A was used as a cell cycle marker and α-tubulin served as a loading control. **d** Quantification of protein levels shown in C (α-tubulin normalized, $n = 2$/experiment, six independent experiments). **e** Effect of CEP85 mutations on PLK4 activation. The PLK4-induced centriole overduplication assays were performed as described in **a**. Selected images showing Myc-PLK4 and PLK4 pS305 labelling. Scale bar 10 μm, white boxes indicate the magnified region. **f, g** The graph indicates the relative levels of Myc-PLK4 and the relative ratio of pS305/PLK4 at centrosomes. ($n = 100$/experiment, three independent experiments). **h** A model for how CEP85 operates in the centriole duplication pathway. CEP85 acts downstream of CEP192, CEP152 and PLK4 in centriole duplication. Direct binding of CEP85 to STIL stabilises STIL and facilitates its recruitment to centrosomes. This further facilitates robust PLK4 activation and subsequent centriole assembly. Two-tailed *t*-test was performed for all *p*-values, all error bars represent SD, and asterisks for *p*-values are \*\**p* < 0.01 and \**p* < 0.05

PLK4, followed by STIL phosphorylation and SASS6 binding to STIL[22, 23, 39, 40]. Our localization data place CEP85 to the proximal end of mother centrioles (Supplementary Fig. 2e). In agreement with Chen et al.[49], we note that the distribution of CEP85 we observe does not display the prototypical ring-like pattern displayed by PLK4. Nevertheless, quantitative immuno-fluorescence measurements of STIL localization during centriole duplication revealed that a fraction of STIL co-localizes with CEP85 early on in the centriole duplication process (Fig. 3a–e), suggesting that the association between CEP85 and STIL occurs transiently at the onset of centriole duplication downstream of CEP192 and upstream of SASS6. Highly dynamic and transient interactions during centriole duplication might occur at other stages as well. Thus, mother centrioles have been proposed to template SASS6 cartwheel assembly in a short-lived transition state before these cartwheels are transported outwards to form daughter centrioles[59].

Our data suggest that the association between CEP85 and STIL is functionally important for full PLK4 activation and efficient centriole duplication. The exact mechanism through which STIL binding by CEP85 translates into PLK4 activation is currently unclear. Binding might be crucial for an efficient local enrichment and/or stabilization of STIL at the place of centriole formation or have a more direct, functional importance for PLK4 activity regulation and/or STIL phosphorylation to facilitate the subsequent steps of the centriole duplication pathway (see model in Fig. 7h).

The conservation of the CEP85-STIL interaction interface implies that this mechanism might be common among metazoans, with the possible exception of insects. A sequence analysis, based on the currently available protein sequences, showed that the STIL NTD and CEP85 are found in metazoan and choanoflagelates but are absent in insects such as *Drosophila* (Supplementary Fig. 8a). In line with this observation is that centriole duplication in *Drosophila* does not always rely on protein–protein interactions found to be critical in other metazoans[60].

CEP85 has previously been shown to interact with NEK2A and to influence centrosome disjunction by antagonising NEK2A activity[49]. While we extend the spectrum of CEP85 activities by showing that its fourth coiled-coil domain interacts with STIL and thereby also plays a role in centriole duplication, the complexities of CEP85's architecture (Fig. 4a) suggest that it might have additional functions in vivo. The cellular functions of STIL are also unlikely to be fully described yet. The structure of the STIL NTD showed the presence of two conserved patches on its surface (Fig. 5a). In the CEP85:STIL complex structure, only one of these patches is involved in engaging CEP85. While the other patch would be well placed to contribute to CEP85 binding (Supplementary Fig. 3c), NMR experiments suggest that this is not the case at least under the condition used in vitro (Supplementary Fig. 3d). Thus, it is possible that the STIL NTD binds other factors, as yet undiscovered, that might compete or synergise with CEP85 binding or be involved in CEP85 independent functions. While STIL regulation during centriole duplication might be subject to additional layers of complexities, our data establish that its interaction with CEP85 contributes a facet to the tight spatiotemporal control of the upstream steps of centriole duplication.

## Methods

Detailed supplementary experimental procedures can be found in Supplementary Information section.

**RNA interference**. For siRNA-mediated depletion of STIL and CEP85, the following oligonucleotide sequences were used: human STIL siRNA[23], 5′-GCUC-CAAACAGUUUCUGCUGGAAU-3′ and human CEP85 siRNA, 5′-

CCUAGAGCAGGAAGUGGCUCAAGAA-3′. siRNAs against human PLK4 (M-005036-02-0005), CEP192 (L-032250-01-0005) and CEP152 (M-022241-01) were purchased from Dharmacon. The Luciferase GL2 Duplex non-targeting siRNA from Dharmacon was used as a negative control. All siRNAs were transfected using Lipofectamine RNAiMax (Invitrogen) according to the manufacturer's protocol.

**Cell assays**. For the S-phase arrest assays (Figs. 2g–j, 6f, g, Supplementary Fig. 7b, c, Supplementary Fig. 7f and Supplementary Fig. 9a-c), cells were transfected with the indicated siRNAs for 72 h. At 48 h post transfection, cells were arrested in S-phase (2 mM hydroxyurea) for 24 h before fixation. For the G2-phase arrest assays (Figs. 1d–f, 6e), cells were transfected with the indicated siRNA for 72 h. At 32 h post transfection, cells were arrested in S-phase (2 mM hydroxyurea) for 24 h and further released into G2 arrest (10 μM RO-3066) for the final 16 h of the treatment. For the PLK4-induced centriole overduplication assays (Figs. 1g, h, k, l, 2a–f and Supplementary Fig. 2f, g), U-2 OS Tet-inducible Myc-PLK4 cells were transfected with the indicated siRNAs for 72 h. At 48 h post transfection, cells were arrested in S-phase (2 mM hydroxyurea) and added tetracycline (2 μg/mL) to induce expression of Myc-PLK4 for 24 h before fixation. For the S-phase arrest induced centriole overduplication assays (Fig. 1i, j), U-2 OS were transfected with the indicated siRNAs for 72 h. At 24 h post transfection, aphidicolin (1.5 μg/mL) was added to arrest cells in S-phase for 48 h before fixation.

**Protein crystallization**. SeMet *T. adhaerens* STIL$^{1-348}$ crystals were obtained using the vapour diffusion method with sitting drops (1 μl of protein solution + 1 μl of reservoir solution) and a reservoir solution of 100 mM MES, pH 6.0, 3% (v/v) MPD at 18 °C. Crystals were mounted after 9 days in 100 mM MES, pH 6.0, 35% (v/v) MPD and frozen in liquid nitrogen.

*H. sapiens* CEP85$^{570-656}$ was crystallized at 18 °C in sitting drops (1 μl of protein solution + 1 μl of reservoir solution) using the vapour diffusion method and a reservoir solution of 100 mM Na-HEPES, pH 7.7, 200 mM Na-Citrate, 36% (v/v) MPD. Crystals were mounted after 6 days and frozen in liquid nitrogen.

The SeMet *T. adhaerens* STIL$^{1-348}$ − *H.sapiens* CEP85$^{570-656}$ complex was crystallized at 18 °C in sitting drops (1 μl of protein solution + 1 μl of reservoir solution) using the vapour diffusion method and a reservoir solution of 100 mM MES, pH 6.0, 1.45 M MgSO$_4$, 2 mM DTT. Crystals were mounted after 2 days in 100 mM MES, pH 6.0, 1.45 M MgSO$_4$, 25% (v/v) glycerol and frozen in liquid nitrogen.

The protein concentrations of the crystallized constructs were determined by the Bradford assay with BSA as a standard and were: 11.0 mg/ml (SeMet *T. adhaerens* STIL$^{1-348}$), 49.9 mg/ml (*H.sapiens* CEP85$^{570-656}$) and 32.8 mg/ml (SeMet *T. adhaerens* STIL$^{1-348}$ + *H.sapiens* CEP85$^{570-656}$).

**Data availability**. The authors declare that all data supporting the findings of this study can be found within the paper and its Supplementary Information files or from the corresponding author upon reasonable request. The BioID mass spectrometry data are available at MassIVE repository (massive.ucsd.edu), accession number MSV000082191. The coordinates and structure factors for the reported X-ray crystallographic structures are deposited in the Protein Data Bank (PDB) under accession codes 5OI7, 5OI9 and 5OID.

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

## Acknowledgements

U-2 OS T-REx cells with Tet-inducible Myc-tagged PLK4 were a kind gift from E. Nigg. We thank Tak W. Mak for the PLK4 pS305 antibody. Y.L. was funded by an Ontario Student Opportunity Trust Fund Award and a Canadian Institutes for Health Research (CIHR) Doctoral Research Award. We thank Robin Buijs for generating U-2 OS STIL CRISPR KO cell line, Mariana Gomez for generating CEP85 constructs, Johnny Tkach for assistance with CRISPR experiments and João Gonçalves for providing the lentiviral GFP expression vector. We acknowledge David Flot (MX1538, ID23-2 and ID29), Daniele DeSanctis (MX1651, ID29), as well as Antoine Royant (MX1651, ID29) at the European Synchrotron Radiation Facility (ESRF), Grenoble, France for their beamline support. We also thank Domagoj Baretic, Alex Berndt and Minmin Yu for their collection of the *T. adhaerens* STIL[1–348] − *H.sapiens* CEP85[570–656] T630M/H652M complex data sets. We are grateful to Mark Allen, MRC-LMB, Cambridge, UK for his kind gift of TEV protease. This work was supported by grants from the Medical Research Council to M.v.B (MRC file reference MC_UP_1201/3) and to S.M. and C.V.R. (MRC Grant No. MR/N020413/1) as well as by the CIHR (MOP-130507 and 142192), the Krembil Foundation and the Ontario Ministry for Research and Innovation RE-8 to L.P.

## Author contributions

Protein purifications and the GST-based pull-down assays were conducted by M.v.B, the structural work, as well as the yeast-two-hybrid experiments by M.v.B. and D.D.B. C.M.J. and S.H.M. performed the biophysical experiments (SEC-MALS, CD, AUC). S.F. conducted the NMR experiments, A.A. did the structural analyses and the evolutionary

bioinformatics. The native mass-spectrometric experiments and cross-linking mass-spectrometric analyses were done by S.M. and C.V.R. and D.W., S.M. and C.V.R., respectively. Y.L. performed most of the cell biological experiments, imaging and in vivo characterisation of the CEP85-STIL complex. Vectors for the STIL MT re-routing assay were made by D.D.B and F.G.A., and the assay was performed by D.D.B. and F.G.A., who also quantified the data. Creation of BioID cell lines, mass spectrometry and analyses were carried out by S.W.T.C., E.C., G.D.G., and B.R., G.D.G. assisted with the quantification of imaging data performed by Y.L. CRISPR-related reagents were prepared by J. M.T. The paper was written by Y.L., L.P. and M.v.B. with contributions from all authors. L.P. and M.v.B. directed the project.

## Additional information

**Competing interests:** The authors declare no competing interests.

