## [Peer Review File · Nature Communications]

Reviewer #1 (Remarks to the Author):

The manuscript by Liu et al describes the identification and characterization of a novel centriolar factor CEP85 in human centriole duplication. The authors provide in-depth structural and biochemical evidence for specific roles of CEP85 in early steps of centriole duplication, particularly during the recruitment of the centriolar protein STIL to the assembly site, where the presence of STIL needs to fully activate the master kinase PLK4 for centriole biogenesis. Guided by crystal structures, the authors found that CEP85 directly interacts with STIL through specific domains/residues that form a conserved interaction interface, and that when the CEP85-STIL interaction is abolished, both the recruitment and stability of STIL are impaired, leading to incomplete activation of PLK4 and thus severe defects in centriole duplication.

I find that the core conclusion of the manuscript regarding the role of CEP85 in centriole duplication is supported by the large amount of data including structural, biochemical and microscopy studies, and that CEP85 is involved in a very important step of centriole duplication previously not recognized. I therefore in principle support the publication of this interesting story in Nature Communication, when the following issue is addressed.

Major issues:

As described by the authors, the co-localization of CEP85 and STIL during centriole duplication is very transient, and it is currently unclear how such transient interactions contribute to or facilitate centriole duplication. I wonder if the authors have considered an alternative idea: It seems to me that CEP85 is profoundly required for the overall stability of STIL in cells, not just at the centrosome only (see Fig 2F & H), and that perhaps the reduction of the total STIL level could underlie the majority of the CEP85 RNAi phenotypes observed. I think the authors should at least discuss this issue, although I guess that one relatively simple way to differentiate the role of CEP85 in STIL stability from that of STIL recruitment is to check if overexpression of STIL can rescue centriole duplication defects in cells depleted of CEP85, i.e. completely bypassing the requirement of CEP85.

Reviewer #2 (Remarks to the Author):

The manuscript "The structural basis for CEP85-mediated control of centriole duplication" by Liu et al reports their identification of CEP85 as a novel

centriole duplication factor directly interacting with the previously uncharacterized N-terminal domain of STIL to spatiotemporally regulate the early stages of centriole duplication. They first identified CEP85 as a new regulator of centriole duplication using BioID with several known centriolar proteins as the baits. Based on a series of RNAi and in vivo data, they found that CEP85 is required for robust accumulation of STIL at centrioles and PLK4 activation. The interaction between CEP85 cc4 and STIL NTD was mapped and confirmed by several biophysical techniques including in vitro pulldown, analytical ultracentrifugation, Y2H, and ITC. Based on the mapped binding sites, they determined the structures of CEP85 cc4 and STIL NTD, first individually to high resolutions of 2.1 and 1.7 Å respectively, and later as a complex to a low resolution of 4.6 Å. Based on the crystal structures, they identified critical residues mediating the interaction which was confirmed by mutagenesis analyses. Interaction-disrupting mutants of CEP cc4 were found to impair both centriole duplication and STIL recruitment to centrosomes, as well as prevent robust activation of PLK4 in vivo. Based on all these findings, they claim the elucidation of the molecular basis underlying a previously undescribed modulatory step during the most upstream events of centriole duplication.

The article was written properly, and the reported finding is novel as they claimed. However, I found some of their data were not so convincing, particularly the complex structure which is the main finding of their work. Listed below are my major and minor concerns.

Major points:

1. Their ITC data show a strong and robust interaction between CEP85 cc4 and STIL NTD. However, in all their in vivo studies, the two proteins only partially (~15%) co-localized, which they concluded as a transient interaction. It is hard to understand how such a strong interaction in vitro leads to only a transient interaction in vivo? Does it imply there is another layer regulation by an unknown mechanism?
2. Their ITC data (Fig 4D) show “N=0.34”, which is inconsistent with their tetrameric structural model of the complex. How to explain this? Similarly, the native MS data (Fig S4D) show that, despite the majority of the tetrameric complex, a substantial fraction of the complex are trimers (2xSTIL:1xCEP85) or even pentamers (3xSTIL:2xCEP85). Further, given the strong dimeric interaction of the coiled coils, it is hard to understand why all CEP85 cc4 are monomers (Fig S4D)? Also looking strange is that most STIL NTD are actually dimers. Overall, it seems very ambiguous how the two proteins behave individually, and whether the crystal structure they determined is really the physiological complex of them in vivo, even they may indeed interact in the cell, transiently as was claimed by the authors.
3. Another concern about the structure of the complex is that it is between *T. adhaerens* STIL NTD, which shares only 28% identity with human ortholog, and human CEP85 cc4. It would be helpful to check the binding of both proteins from the same organism, e.g. *T. adhaerens* STIL vs *T. adhaerens* CEP85, by ITC to confirm the interactions are comparable in both cases. Along the same line, crosslinking-MS analyses would help to further validate the interaction sites given the very low resolution of their complex structure.

4. Regarding the unwinding of the N-terminal part of CEP85 cc4 upon binding of STIL NTD, more discussion of the cause and potential function should be considered. Could this be tested by some techniques such as limited proteolysis or NMR? Does a short version of CEP85 cc4 corresponding to the seen part in the complex structure bind in the same manner (and with a similar affinity) as the one used in their studies? If so, SAXS might be used to further confirm the tetrameric complex.

Minor points:

1. "Conservation Scale" bars were shown several times (Fig 5A, Fig S3C, Fig S7D). However, it was unclear how the values were calculated? Further, more homologous proteins should be included in their alignments (Fig S7B&C) to demonstrate residue conservations.
2. How to explain that depletion of CEP85 reduced STIL level, but depletion of STIL rather increased CEP85 level (Fig 2D)?
3. Regarding the reasoning of absence of CEP85 in nematodes and flies (Fig S7A), it would be helpful to compare the structure of the STIL NTD reported here with that of the counterpart in Sas5/Ana2.

Reviewer #3 (Remarks to the Author):

The manuscript by Liu et al describes the identification the protein CEP85 as a new player in centriole duplication. Initially, the protein CEP85 is identified to be interacting with known centriole duplication factors by a set of protein proximity detection methods, and subsequently, its role in the centriole duplication process is established and validated in vivo. Two high resolution structures, one of the interacting domains CEP85-CC4 and one of its binding partner STIL-NTD are determined, and a low resolution structure of the protein-protein complex is presented that fits well to the evolutionary conservation of these domain. The interaction between CEP85 and STIL is further characterized by biophysical methods including NMR spectroscopy. Single-point mutations, based on the crystal structures are analyzed in vitro and in vivo, demonstrating that the CEP85-STIL interaction is essential for STIL localization to centrioles and subsequent PLK4 activation and finally correct daughter centriole formation.

The work comprises an overall impressive amount of data and experiments, which appear all very well done and well documented. The amount of work shown here is clearly above average and fully sufficient to warrant publication. The manuscript is clearly written and the provided findings

represent a major scientific step forward in the centriole field. I highly recommend publication in Nature communications. Two minor issues may be addressed:

1.) The title "Structural basis of .." is in my view even too modest and includes only part of the achievements that the manuscript provides. Perhaps the authors want to change the title to a more general term to highlight that they also identify the role of CEP85 in the first place and elucidate its function in vivo at least partially.

2.) The authors conclude from the absence of a detectable interaction with the R67A mutant in NMR and ITC experiments (Fig. S3), that CEP85 does not interact with STIL via its patch2. This conclusion should be worded and discussed more carefully. The only statement that can be made safely is that under the conditions used in vitro, this interaction not detected. It may nonetheless exist in vivo, perhaps with a phosphorylation or other regulation or under otherwise different conditions.

We would like to thank the three referees for their enthusiasm about our work, judicious comments and the many thoughtful suggestions. Below is our detailed point-by point response (our responses are **in bold** and the original comments in their entirety are *in italics*). The referee reports have been very helpful and we hope that the reviewers will find the revised version of our manuscript suitable for publication in *Nature Communications*.

Reviewer #1:

The manuscript by Liu et al describes the identification and characterization of a novel centriolar factor CEP85 in human centriole duplication. The authors provide in-depth structural and biochemical evidence for specific roles of CEP85 in early steps of centriole duplication, particularly during the recruitment of the centriolar protein STIL to the assembly site, where the presence of STIL needs to fully activate the master kinase PLK4 for centriole biogenesis. Guided by crystal structures, the authors found that CEP85 directly interacts with STIL through specific domains/residues that form a conserved interaction interface, and that when the CEP85-STIL interaction is abolished, both the recruitment and stability of STIL are impaired, leading to incomplete activation of PLK4 and thus severe defects in centriole duplication.

I find that the core conclusion of the manuscript regarding the role of CEP85 in centriole duplication is supported by the large amount of data including structural, biochemical and microscopy studies, and that CEP85 is involved in a very important step of centriole duplication previously not recognized. I therefore in principle support the publication of this interesting story in Nature Communication, when the following issue is addressed.

Major issues:

As described by the authors, the co-localization of CEP85 and STIL during centriole duplication is very transient, and it is currently unclear how such transient interactions contribute to or facilitate centriole duplication. I wonder if the authors have considered an alternative idea: It seems to me that CEP85 is profoundly required for the overall stability of STIL in cells, not just at the centrosome only (see Fig 2F & H), and that perhaps the reduction of the total STIL level could underlie the majority of the CEP85 RNAi phenotypes observed. I think the authors should at least discuss this issue, although I guess that one relatively simple way to differentiate the role of CEP85 in STIL stability from that of STIL recruitment is to check if overexpression of STIL can rescue centriole duplication defects in cells depleted of CEP85, i.e. completely bypassing the requirement of CEP85.

We thank the reviewer for her/his thoughtful comments. As suggested, we performed rescue experiments where we overexpress STIL in CEP85-depleted cells, to assess the level of centriole duplication in S-phase. Our results indicate that expression of WT STIL as well as STIL L64A and R67A mutant are unable to rescue centriole duplication,

supporting the role of CEP85 in facilitating STIL recruitment to centrioles (Figure S8A-C). To further validate this point, we overexpressed a non-degradable form of WT STIL and STIL L64A and R67A mutant to assess their ability in centriole amplification¹. Consistently, we found that expression of similar level of STIL L64A and R67A mutant was unable to induce centriole overduplication to WT STIL levels (Figure S8D-F). Together, our data support a dual role for CEP85 in STIL centriolar localization and its stability to control centriole duplication.

This comment from the reviewer also made us realize that it was necessary to investigate which factors are required for the recruitment of CEP85 to centrioles. To do this, we depleted CEP192, CEP152, PLK4 and STIL in U-2 OS cells and examined the impact on CEP85 centriolar localization. We found that depletion of CEP192, CEP152 and PLK4 led to a reduction in centriolar recruitment of CEP85 (Figure S2F-G). These data suggest that CEP85 acts downstream of CEP192, CEP152, and PLK4, and therefore the model in Figure 7H has been modified accordingly. These results are mentioned on page 8 of the revised manuscript.

Reviewer #2:

The manuscript “The structural basis for CEP85-mediated control of centriole duplication” by Liu et al reports their identification of CEP85 as a novel centriole duplication factor directly interacting with the previously uncharacterized N-terminal domain of STIL to spatiotemporally regulate the early stages of centriole duplication. They first identified CEP85 as a new regulator of centriole duplication using BioID with several known centriolar proteins as the baits. Based on a series of RNAi and in vivo data, they found that CEP85 is required for robust accumulation of STIL at centrioles and PLK4 activation. The interaction between CEP85 cc4 and STIL NTD was mapped and confirmed by several biophysical techniques including in vitro pulldown, analytical ultracentrifugation, Y2H, and ITC. Based on the mapped binding sites, they determined the structures of CEP85 cc4 and STIL NTD, first individually to high resolutions of 2.1 and 1.7 Å respectively, and later as a complex to a low resolution of 4.6 Å. Based on the crystal structures, they identified critical residues mediating the interaction which was confirmed by mutagenesis analyses. Interaction-disrupting mutants of CEP cc4 were found to impair both centriole duplication and STIL recruitment to centrosomes, as well as prevent robust activation of PLK4 in vivo. Based on all these findings, they claim the elucidation of the molecular basis underlying a previously undescribed modulatory step during the most upstream events of centriole duplication. The article was written properly, and the reported finding is novel as they claimed. However, I found some of their data were not so convincing, particularly the complex structure which is the main finding of their work. Listed below are my major and minor concerns.

Major points:

1. Their ITC data show a strong and robust interaction between CEP85 cc4 and STIL NTD.

However, in all their in vivo studies, the two proteins only partially (~15%) co-localized, which they concluded as a transient interaction. It is hard to understand how such a strong interaction in vitro leads to only a transient interaction in vivo? Does it imply there is another layer of regulation by an unknown mechanism?

The ITC/AUC experiments that we had presented in the manuscript suggested that the CEP85-STIL binding affinity is ~ 20 μ M. Additional ITC and AUC experiments designed to clarify the binding stoichiometry of the complex (see comments below, Figure S4C, Figure S5G) suggest a K_D of ~ 60 μ M. These values classify this interaction as a relatively weak interaction, which might partially explain the putatively transient interaction at centrosomes observed in vivo.

The complex might also play a role in the cytoplasm. Holland and colleagues propose that cytoplasmic STIL needs to associate with PLK4 to transform into a functional conformation in order to be recruited to centrioles². So, in analogy, we think that CEP85 may interact with both centrosomal and cytoplasmic pools of STIL to play its dual regulation on STIL. In agreement with this notion, our microtubule recruitment assay in Figure 6H-I indicate CEP85 can robustly recruit cytoplasmic STIL to microtubules. This is now also discussed in the manuscript. However, the reviewer is of course right in pointing out that there might indeed be further layers of regulation by unknown mechanisms that remain to be explored.

2. Their ITC data (Fig 4D) show “N=0.34”, which is inconsistent with their tetrameric structural model of the complex. How to explain this? Similarly, the native MS data (Fig S4D) show that, despite the majority of the tetrameric complex, a substantial fraction of the complex are trimers (2xSTIL:1xCEP85) or even pentamers (3xSTIL:2xCEP85). Further, given the strong dimeric interaction of the coiled coils, it is hard to understand why all CEP85 cc4 are monomers (Fig S4D)? Also looking strange is that most STIL NTD are actually dimers. Overall, it seems very ambiguous how the two proteins behave individually, and whether the crystal structure they determined is really the physiological complex of them in vivo, even they may indeed interact in the cell, transiently as was claimed by the authors.

Concerning the ITC binding stoichiometry: We had demonstrated by MALS that the CEP85 cc4 is in fact partially unstable at room temperature (Figure S5B). We have now also performed thermal melts which emphasizes this point further (Figure S5F). This instability probably explains the lower binding stoichiometry observed at room temperature compared to our structure (Figure S5G, shortly discussed in the corresponding figure legend). However, to address the point of binding stoichiometry directly, we have now done ITC experiments at reduced temperature (10°C) to stabilise the CEP85 cc4 (Figure S5G) and also redone our AUC experiments under conditions optimised to resolve the binding stoichiometry of the complex (Figure S4C). Both experiments demonstrate a binding stoichiometry of 1:1 (2:2), which is in perfect agreement with our

structural data (the AUC data resolved both 1:2 and 2:2 complexes). In further support of the idea that the stability of cc4 somewhat compromises the observed ITC stoichiometry at 25 °C, we have now also performed binding experiments by ITC at 25 °C using a longer construct of CEP85. This construct contains an additional coiled coil element (cc5) and is more stable than WT CEP85 cc4 alone as judged by thermal melts (Figure 1B for the reviewers' attention). With this construct we observed robust binding to STIL NTD of comparable affinity and binding stoichiometry of 1:1 (2:2), as observed for CEP85 cc4 at 10°C (Figure 1A for the reviewers' attention).

Second, concerning the apparent ambiguous behaviour of the proteins individually. These are only observed in native mass-spectrometry experiments, where the proteins are injected at high concentrations (due to the relatively low K_D) and run in vacuum (removing the solvation shell of water). Under milder conditions, in solution and at lower concentrations, such as MALS (Figure S5A+B) at room temperature, the CEP85 cc4 is in monomer-dimer equilibrium and the STIL NTD is predominantly monomeric. To address this point further, we also have performed AUC at reduced temperature (10 °C) with the individual proteins and find this notion (CEP85 cc dimer, STIL NTD monomer) confirmed. Thus, we believe that the different behaviour of the individual proteins observed in native mass-spectrometry might be down to the experimental conditions. The native mass-spectrometry data nevertheless confirms complex formation and also confirms that the CEP85 cc4 binds only as a dimer to the STIL NTD. It is therefore in agreement with our conclusions concerning CEP85-STIL binding.

3. Another concern about the structure of the complex is that it is between T. adhaerens STIL NTD, which shares only 28% identity with human ortholog, and human CEP85 cc4. It would be helpful to check the binding of both proteins from the same organism, e.g. T. adhaerens STIL vs T. adhaerens CEP85, by ITC to confirm the interactions are comparable in both cases. Along the same line, crosslinking-MS analyses would help to further validate the interaction sites given the very low resolution of their complex structure.

We had shown that point mutations in the interface of our structure strongly compromise the binding between the human proteins (based on pull-downs and a microtubule-based recruitment assay *in vivo*) as well as between chicken STIL NTD and human CEP85 cc4 in ITC experiments with recombinant proteins. These experiments argue that, despite being obtained with *Trichoplax* STIL NTD and human CEP85 cc4, our structure is nevertheless relevant. To strengthen this point further we have now also used cross-linking MS analyses, as suggested by the reviewer. This experiment demonstrates the presence of a specific cross-link between chicken STIL NTD and human CEP85 cc4 for the WT, but not the mutant proteins. This cross-link is in agreement with our structural model (Figure S5H-K).

In addition, we have now also conducted ITC experiments with human STIL NTD vs human CEP85 cc4 and chicken STIL NTD vs chicken CEP85 cc4 that show binding of the WT, but not the mutant proteins (Figure 1C+D for the reviewers' attention). Fitting the WT data to obtain the K_D and binding stoichiometry of these intra-species ITC data was

difficult because the parameters were poorly constrained by the fit under the experimental conditions that we could access. Both human and chicken interactions may have lower binding affinities and, in the case of human STIL NTD, we were unable to obtain sufficiently concentrated stocks without evidence of aggregation. In the case of the chicken STIL NTD – human Cep85 cc4 experiments we required >1 mM STIL NTD stock (>41 mg/ml) and we were unable to duplicate this with the human construct. The chicken interaction may also occur with a more complex binding mode, for example possibly involving the conserved patch 2 of STIL NTD, or may require further optimisation of solvent conditions since the chicken cc4 sequence contains two cysteine residues that are well placed in the parallel cc dimer to form disulphide linked material, despite the presence of DTT in the buffer. Although we are reluctant to fit these data quantitatively, it is clear that the proteins from both species bind to each other and that the same mutants in the conserved binding interface that disrupt the chicken – human interaction also prevent binding for the corresponding intra-species interactions. All of this is in complete agreement with our structural model.

We also attempted ITC experiments at 10 °C using *Trichoplax* STIL NTD vs *Trichoplax* CEP85 cc4 but without success. However, SEC-MALS and CD experiments showed that the *Trichoplax* CEP85 cc4 was already partially unfolded even at 4 °C and essentially monomeric at room temperature (Figure 1E+F for the reviewers' attention). Since CEP85 cc4 needs to be dimeric to bind to STIL NTD (Figure S4D), this instability compared to human (Figure S5B, Figure S5F) and chicken CEP85 cc4 (Figure 1F for the reviewers' attention) likely explains the lack of a robust interaction in ITC experiments. CEP85 contains a number of additional coiled coil domains (see Figure 4A for a domain overview of CEP85) that might act to stabilise cc4 dimer formation in the context of full length CEP85. Indeed, using human CEP85, we found that in the absence of cc4, CEP85 is still able to oligomerise (Figure 1G for the reviewers' attention).

4. Regarding the unwinding of the N-terminal part of CEP85 cc4 upon binding of STIL NTD, more discussion of the cause and potential function should be considered. Could this be tested by some techniques such as limited proteolysis or NMR? Does a short version of CEP85 cc4 corresponding to the seen part in the complex structure bind in the same manner (and with a similar affinity) as the one used in their studies? If so, SAXS might be used to further confirm the tetrameric complex.

We had tried in the past to shorten the CEP85 cc4 further, but did not see any binding to STIL NTD. This is not unexpected, since CEP85 cc4 can only bind as a dimer to STIL (Figure S4D) and taking off more heptad repeats from the coiled coil would act to destabilise dimer formation (the coiled coil is already partly unstable at room temperature (Figure S5B+F)).

In our opinion, it would be very challenging to use limited proteolysis to check the fraying of the CEP85 cc4 in solution and when bound to STIL. The binding affinities are relatively weak making it difficult to obtain unique complexes and therefore the nature of the relevant controls is not clear. Thus, any result would be difficult to interpret unambiguously. SAXS under the required high protein concentrations (relatively low K_D) would also lead to technical difficulties (concentration effects, multiple states, aggregation etc.), besides being unlikely to be able to differentiate between a fully folded or partially unfolded N-terminal coiled-coil part in the complex (SAXS is a low resolution technique). NMR experiments would require a full assignment of the CEP85 cc4 dimer, which is beyond the time-scale of this revision (and, due to the nature of the parallel coiled coil dimer, being non-trivial). Thus, unfortunately, we feel that we are unable to address this point experimentally. However, partial unwinding of proteins to enable crystal packing is not uncommon in protein crystallography. Thus, as discussed in the manuscript, we believe that this is the most likely explanation in our case as well, especially given the partial instability of the CEP85 cc4 at room temperature.

Minor points:

1. “Conservation Scale” bars were shown several times (Fig 5A, Fig S3C, Fig S7D). However, it was unclear how the values were calculated? Further, more homologous proteins should be included in their alignments (Fig S7B&C) to demonstrate residue conservations.

The conservation scores for both CEP85 cc4 and STIL NTD, were calculated with ConSurf using manually refined multiple sequence alignments each of which contained 136 non-redundant homologous sequences from the same set of species. The position-specific scores were calculated using a Bayesian algorithm. These scores are divided into a discrete scale of nine grades and indicate the relative degree of evolutionary conservation at each amino acid position in the given alignment. We integrated this information into the Materials and Methods.

For an extended multiple sequence alignment please refer to Figure 2 for the reviewers’ attention. The extended alignment for both CEP85 cc4 and STIL NTD includes sequences from diverse metazoan organisms representing the main branches of the phylogenetic tree shown in Figure S7. It clearly shows that the residues mutated in our study are highly conserved across species. Similarly, the regions (blocks) corresponding to the secondary structural elements are well conserved. The number of Supplementary Figures is limited and these alignments are very bulky without, to our mind, carrying extra or essential information. Thus, we would prefer to retain in the manuscript supplement their shorter version that includes the organisms used in our study.

2. How to explain that depletion of CEP85 reduced STIL level, but depletion of STIL rather increased CEP85 level (Fig 2D)?

This is a very good question and we must confess that this is also a puzzling observation for which we have no concrete explanation. Our data clearly indicate that CEP85-STIL can form a complex *in vitro*, and we think a plausible explanation is that the regulated stability of this complex (or its individual components) *in vivo* may be a significant factor determining its bioavailability. We have shown that CEP85 acts upstream of STIL in centriole duplication. In the absence of CEP85, STIL is unable to fulfill its physiological functions and therefore may promote its degradation by specific E3 ubiquitin ligases. This is not without precedent since other centriole duplication factors^{1,3-6} (SASS6, CPAP) have been shown to regulate cellular levels of core duplication factors. Consistent with this observation in Figure 2D, our new data indicate that STIL depletion also increased the centriolar level of CEP85 (Figure S2F-G). Those data imply a potential feedback regulation of CEP85 levels that warrants further investigation. We previously reported a similar phenomenon that depletion of CEP120, SPICE1, CPAP and CEP135 resulted in a marked increase in the PLK4 signal surrounding the mother centriole⁷. Further work is needed to identify UPS components that potentially regulate the CEP85/STIL complex.

3. Regarding the reasoning of absence of CEP85 in nematodes and flies (Fig S7A), it would be helpful to compare the structure of the STIL NTD reported here with that of the counterpart in Sas5/Ana2.

Intriguingly, the fly and nematode homologs of STIL, Ana2 and Sas5, both do not have a NTD (schematically shown in Figure S7). In fact, our sequence analysis revealed that insects such as wasps, ants, butterflies, beetles and bees also lack the STIL NTD. We were unable to identify a CEP85 homologue in these organisms, which could be taken as a further indication that the described interaction between CEP85 and STIL is evolutionarily relevant. A small paragraph in the discussion of the manuscript describes our findings (also see Figure S7A). As discussed in the legend of Figure S7, conclusions concerning nematode SAS-5 are difficult though, since its homology to STIL is not apparent from sequence comparison.

Reviewer #3

The manuscript by Liu et al describes the identification the protein CEP85 as a new player in centriole duplication. Initially, the protein CEP85 is identified to be interacting with known centriole duplication factors by a set of protein proximity detection methods, and subsequently, its role in the centriole duplication process is established and validated in vivo. Two high resolution structures, one of the interacting domains CEP85-CC4 and one of its binding partner STIL-NTD are determined, and a low resolution structure of the protein-protein complex is presented that fits well to the evolutionary conservation of these domain. The interaction between CEP85 and STIL is further characterized by biophysical methods including NMR spectroscopy. Single-point mutations, based on the crystal structures are analyzed in vitro and in vivo, demonstrating that the CEP85-STIL interaction is essential for STIL localization to centrioles and subsequent PLK4 activation and finally correct

daughter centriole formation. The work comprises an overall impressive amount of data and experiments, which appear all very well done and well documented. The amount of work shown here is clearly above average and fully sufficient to warrant publication. The manuscript is clearly written and the provided findings represent a major scientific step forward in the centriole field. I highly recommend publication in Nature communications. Two minor issues may be addressed:

We thank this reviewer for her/his enthusiasm.

1.) The title "Structural basis of .." is in my view even too modest and includes only part of the achievements that the manuscript provides. Perhaps the authors want to change the title to a more general term to highlight that they also identify the role of CEP85 in the first place and elucidate its function in vivo at least partially.

We appreciate this kind suggestion. We changed the title to “Direct binding of CEP85 to STIL ensures robust PLK4 activation and efficient centriole assembly”.

2.) The authors conclude from the absence of a detectable interaction with the R67A mutant in NMR and ITC experiments (Fig. S3), that CEP85 does not interact with STIL via its patch2. This conclusion should be worded and discussed more carefully. The only statement that can be made safely is that under the conditions used in vitro, this interaction not detected. It may nonetheless exist in vivo, perhaps with a phosphorylation or other regulation or under otherwise different conditions.

This is a good point. We now mentioned on Page 22 of the manuscript “While the other patch would be well placed to contribute to CEP85 binding (Figure S3C), NMR experiments suggest that this is not the case at least under the condition used in vitro (Figure S3D)”

References

1. Arquint, C. & Nigg, Erich A. STIL microcephaly mutations interfere with APC/C-mediated degradation and cause centriole amplification. *Current Biology* **24**, 351-360 (2014).
2. Moyer, T.C., Clutario, K.M., Lambrus, B.G., Daggubati, V. & Holland, A.J. Binding of STIL to Plk4 activates kinase activity to promote centriole assembly. *The Journal of Cell Biology* **209**, 863-878 (2015).
3. Puklowski, A. *et al.* The SCF–FBXW5 E3-ubiquitin ligase is regulated by PLK4 and targets HsSAS-6 to control centrosome duplication. *Nature Cell Biology* **13**, 1004 (2011).
4. Rogers, G.C., Rusan, N.M., Roberts, D.M., Peifer, M. & Rogers, S.L. The SCF Slimb ubiquitin ligase regulates Plk4/Sak levels to block centriole reduplication. *The Journal of Cell Biology* **184**, 225-239 (2009).
5. Cunha-Ferreira, I. *et al.* The SCF/Slimb Ubiquitin Ligase Limits Centrosome Amplification through Degradation of SAK/PLK4. *Current Biology* **19**, 43-49 (2009).
6. Kim, M.K., Dudognon, C. & Smith, S. Tankyrase 1 regulates centrosome function by controlling CPAP stability. *EMBO Reports* **13**, 724-732 (2012).
7. Comartin, D. *et al.* CEP120 and SPICE1 cooperate with CPAP in centriole elongation. *Current Biology* **23**, 1360-1366 (2013).

Figure 1 for reviewers

(A, B). Recombinant chicken STIL NTD and the human C-terminal CEP85 construct CEP85⁵⁶²⁻⁷⁶² (including CEP85 cc4 and cc5) directly interact with each other at 25 °C. Binding affinity and stoichiometry are comparable to ITC experiments with human CEP85 cc4 at 10 °C (Figure S5G). At 10 °C cc4 dimer formation is stabilised (Figure S5F). (A). ITC binding isotherm for chicken STIL NTD titrated into human CEP85⁵⁶²⁻⁷⁶² at 25 °C. The resulting K_D , ΔH and STIL NTD/ CEP85⁵⁶²⁻⁷⁶² binding stoichiometry (N) as an average from a total of three measurements are indicated. (B). CD-based thermal melting analysis of recombinant human CEP85⁵⁷⁰⁻⁶⁶² cc4 and CEP85⁵⁶²⁻⁷⁶². Note the increased thermal stability of CEP85⁵⁶²⁻⁷⁶² compared to CEP85⁵⁷⁰⁻⁶⁶² cc4. CEP85⁵⁶²⁻⁷⁶² contains an additional coiled coil domain (cc5, Figure 4A). (C, D). Recombinant STIL NTD and CEP85 cc4 from different species directly interact with each through a conserved interface. (C). ITC binding isotherm for human STIL NTD (WT and R67A mutant) titrated into human CEP85 cc4 (WT and Q640A) at 10 °C. (D). ITC binding isotherm for chicken STIL NTD (WT and R63A mutant (R67A in human STIL) titrated into chicken CEP85 cc4 (WT and Q659A mutant (Q640A in human CEP85) at 10°C. At this temperature, these interactions are driven by a large favourable entropy of binding similar to what is observed with the chicken STIL NTD – hs CEP85 cc4 titrations (Figure S5G). Please note that it was not possible to fit the data from (C, D). unambiguously to obtain K_D and binding stoichiometries, due to the relevant parameters being poorly constrained by the fit. This could be due to several reasons, e.g., in the case of the human STIL NTD, an aggregation tendency, somewhat lower binding affinities, or more complex binding modes that might involve additional interaction sites (e.g. conserved patch 2 of STIL NTD). (E, F). Recombinant *Trichoplax* CEP85 cc4 does not form stable dimers in solution. (E). Size exclusion chromatography - multi-angle light scattering (SEC-MALS) chromatograms of recombinant *Trichoplax* CEP85 cc4 run at room temperature at varying concentrations. Shown are the respective refractive index signals together with the derived molar masses (indicated by thicker horizontal lines). The calculated, theoretical molecular weight is indicated. *Trichoplax* CEP85 cc4 remained monomeric over the concentration range examined (the average molecular weight in the indicated regions ranged from 13-15 kDa). (F). CD-based thermal melting analysis of recombinant *Trichoplax* and *Gallus* CEP85 cc4 both at 0.6 mg/ml. Note the low thermal stability of the *Trichoplax* coiled coil. (G). CEP85 contains additional oligomerisation domains besides its cc4 domain. 293T cells expressing Tet-inducible FLAG-BirA* or FLAG-BirA* tagged human CEP85 WT transgenes were transfected with MYC-CEP85 WT or cc4 deletion constructs for 48 h in the presence of tetracycline (2µg/mL), and immunoprecipitated using FLAG antibody-conjugated beads. FLAG-BirA* CEP85 and co-immunoprecipitated proteins were probed with the indicated antibodies.

Figure 2 for reviewers

The STIL NTD/CEP85 interaction interface is conserved across metazoans. Extended multiple sequence alignment of STIL NTD (A) or CEP85 cc4 (B) homologues from diverse metazoan organisms, representing the major branches of the tree shown in Figure S7. The alignments are coloured according to the CLUSTAL coloring scheme, residue color intensity is based on conservation. The secondary structure elements are

shown above the alignment. The two STIL / CEP85 residues mutated in this study are indicated with red dots.

Figure 1 for reviewers

Reviewer #1 (Remarks to the Author):

The authors have fully addressed the issues I raised. The story is interesting and important and I therefore fully support its publication in nature communication.

Reviewer #2 (Remarks to the Author):

My comments and concerns have been adequately addressed. In my opinion, it is ready for publication.

Reviewers' comments:

Reviewer #1 (Remarks to the author):

The manuscript by Liu et al describes the identification and characterization of a novel centriolar factor CEP85 in human centriole duplication. The authors provide in-depth structural and biochemical evidence for specific roles of CEP85 in early steps of centriole duplication, particularly during the recruitment of the centriolar protein STIL to the assembly site, where the presence of STIL needs to fully activate the master kinase PLK4 for centriole biogenesis. Guided by crystal structures, the authors found that CEP85 directly interacts with STIL through specific domains/residues that form a conserved interaction interface, and that when the CEP85-STIL interaction is abolished, both the recruitment and stability of STIL are impaired, leading to incomplete activation of PLK4 and thus severe defects in centriole duplication.

I find that the core conclusion of the manuscript regarding the role of CEP85 in centriole duplication is supported by the large amount of data including structural, biochemical and microscopy studies, and that CEP85 is involved in a very important step of centriole duplication previously not recognized. I therefore in principle support the publication of this interesting story in Nature Communication, when the following issue is addressed.

Major issues:

As described by the authors, the co-localization of CEP85 and STIL during centriole duplication is very transient, and it is currently unclear how such transient interactions contribute to or facilitate centriole duplication. I wonder if the authors have considered an alternative idea: It seems to me that CEP85 is profoundly required for the overall stability of STIL in cells, not just at the centrosome only (see Fig 2F & H), and that perhaps the reduction of the total STIL level could underlie the majority of the CEP85 RNAi phenotypes observed. I think the authors should at least discuss this issue, although I guess that one relatively simple way to differentiate the role of CEP85 in STIL stability from that of STIL recruitment is to check if overexpression of STIL can rescue centriole duplication defects in cells depleted of CEP85, i.e. completely bypassing the requirement of CEP85.

Reviewer #2 (Remarks to the author):

The manuscript “The structural basis for CEP85-mediated control of centriole duplication” by Liu et al reports their identification of CEP85 as a novel centriole duplication factor directly interacting with the previously uncharacterized N-terminal domain of STIL to spatiotemporally regulate the early stages of centriole duplication. They first identified CEP85 as a new regulator of centriole duplication using BioID with several known centriolar proteins as the baits. Based on a series of RNAi and in vivo data, they found that CEP85 is required for robust accumulation of STIL at centrioles and PLK4 activation. The interaction between CEP85 cc4 and STIL NTD was mapped and confirmed by several biophysical techniques including in vitro pulldown, analytical ultracentrifugation, Y2H, and ITC. Based on the mapped binding sites, they determined the structures of CEP85 cc4 and STIL NTD, first individually to high resolutions of 2.1 and 1.7 Å respectively, and later as a complex to a low resolution of 4.6 Å. Based on the crystal structures, they identified critical residues mediating the interaction which was confirmed by mutagenesis analyses. Interaction-disrupting mutants of CEP cc4 were found to impair both centriole duplication and STIL recruitment to centrosomes, as well as prevent robust activation of PLK4 in vivo. Based on all these findings, they claim the elucidation of the molecular basis underlying a previously undescribed modulatory step during the most upstream events of centriole duplication. The article was written properly, and the reported finding is novel as they claimed. However, I found some of their data were not so convincing, particularly the complex structure which is the main finding of their work. Listed below are my major and minor concerns.

Major points:

1. Their ITC data show a strong and robust interaction between CEP85 cc4 and STIL NTD. However, in all their in vivo studies, the two proteins only partially (~15%) co-localized, which they concluded as a transient interaction. It is hard to understand how such a strong interaction in vitro leads to only a transient interaction in vivo? Does it imply there is another layer regulation by an unknown mechanism?

2. Their ITC data (Fig 4D) show “N=0.34”, which is inconsistent with their tetrameric structural model of the complex. How to explain this? Similarly, the native MS data (Fig S4D) show that, despite the majority of the tetrameric complex, a substantial fraction of the complex are trimers (2xSTIL:1xCEP85) or even pentamers (3xSTIL:2xCEP85). Further, given the strong dimeric interaction of the coiled coils, it is hard to understand why all CEP85 cc4 are monomers (Fig S4D)? Also looking strange is that most STIL NTD are actually dimers. Overall, it seems very ambiguous how the two proteins behave individually, and whether the crystal structure they determined is really the physiological complex of them in vivo, even they may indeed interact in the cell, transiently as was claimed by the authors.

3. Another concern about the structure of the complex is that it is between *T. adhaerens* STIL NTD, which shares only 28% identity with human ortholog, and human CEP85 cc4. It would be helpful to check the binding of both proteins from the same organism, e.g. *T. adhaerens* STIL vs *T. adhaerens* CEP85, by ITC to confirm the interactions are comparable in both cases. Along the same line, crosslinking-MS analyses would help to further validate the interaction sites given the very low resolution of their complex structure.

4. Regarding the unwinding of the N-terminal part of CEP85 cc4 upon binding of STIL NTD, more discussion of the cause and potential function should be considered. Could this be tested by some techniques such as limited proteolysis or NMR? Does a short version of CEP85 cc4 corresponding to the seen part in the complex structure bind in the same manner (and with a similar affinity) as the one used in their studies? If so, SAXS might be used to further confirm the tetrameric complex.

Minor points:

1. “Conservation Scale” bars were shown several times (Fig 5A, Fig S3C, Fig S7D). However, it was unclear how the values were calculated? Further, more homologous proteins should be included in their alignments (Fig S7B&C) to demonstrate residue conservations.

2. How to explain that depletion of CEP85 reduced STIL level, but depletion of STIL rather increased CEP85 level (Fig 2D)?

3. Regarding the reasoning of absence of CEP85 in nematodes and flies (Fig S7A), it would be helpful to compare the structure of the STIL NTD reported here with that of the counterpart in *Sas5/Ana2*.

Reviewer #3 (Remarks to the author):

The manuscript by Liu et al describes the identification the protein CEP85 as a new player in centriole duplication. Initially, the protein CEP85 is identified to be interacting with known centriole duplication factors by a set of protein proximity detection methods, and subsequently, its role in the centriole duplication process is established and validated in vivo. Two high resolution structures, one of the interacting domains CEP85-CC4 and one of its binding partner STIL-NTD are determined, and a low resolution structure of the protein-protein complex is presented that fits well to the evolutionary conservation of these domain. The interaction between CEP85 and STIL is further characterized by biophysical methods including NMR spectroscopy. Single-point mutations, based on the crystal structures are analyzed in vitro and in vivo, demonstrating that the CEP85-STIL interaction is essential for STIL localization to centrioles and subsequent PLK4 activation and finally correct daughter centriole formation. The work comprises an overall impressive amount of data and experiments, which appear all very well done and well documented. The amount of work shown here is clearly above average and fully sufficient to warrant publication. The manuscript is clearly written and the provided findings represent a major scientific step forward in the centriole field. I highly recommend publication in Nature communications. Two minor issues may be addressed:

- 1.) The title "Structural basis of .." is in my view even too modest and includes only part of the achievements that the manuscript provides. Perhaps the authors want to change the title to a more general term to highlight that they also identify the role of CEP85 in the first place and elucidate its function in vivo at least partially.
- 2.) The authors conclude from the absence of a detectable interaction with the R67A mutant in NMR and ITC experiments (Fig. S3), that CEP85 does not interact with STIL via its patch2. This conclusion should be worded and discussed more carefully. The only statement that can be made safely is that under the conditions used in vitro, this interaction not detected. It may nonetheless exist in vivo, perhaps with a phosphorylation or other regulation or under otherwise different conditions.

Responses to the reviewers

We would like to thank the three referees for their enthusiasm about our work, judicious comments and the many thoughtful suggestions. Below is our detailed point-by point response (our responses are in **bold** and the original comments in their entirety are in italics). The referee reports have been very helpful and we hope that the reviewers will find the revised version of our manuscript suitable for publication in Nature Communications.

Reviewer #1:

The manuscript by Liu et al describes the identification and characterization of a novel centriolar factor CEP85 in human centriole duplication. The authors provide in-depth structural and biochemical evidence for specific roles of CEP85 in early steps of centriole duplication, particularly during the recruitment of the centriolar protein STIL to the assembly site, where the presence of STIL needs to fully activate the master kinase PLK4 for centriole biogenesis. Guided by crystal structures, the authors found that CEP85 directly interacts with STIL through specific domains/residues that form a conserved interaction interface, and that when the CEP85-STIL interaction is abolished, both the recruitment and stability of STIL are impaired, leading to incomplete activation of PLK4 and thus severe defects in centriole duplication.

I find that the core conclusion of the manuscript regarding the role of CEP85 in centriole duplication is supported by the large amount of data including structural, biochemical and microscopy studies, and that CEP85 is involved in a very important step of centriole duplication previously not recognized. I therefore in principle support the publication of this interesting story in Nature Communication, when the following issue is addressed.

Major issues:

As described by the authors, the co-localization of CEP85 and STIL during centriole duplication is very transient, and it is currently unclear how such transient interactions contribute to or facilitate centriole duplication. I wonder if the authors have considered an alternative idea: It seems to me that CEP85 is profoundly required for the overall stability of STIL in cells, not just at the centrosome only (see Fig 2F & H), and that perhaps the reduction of the total STIL level could underlie the majority of the CEP85 RNAi phenotypes observed. I think the authors should at least discuss this issue, although I guess that one relatively simple way to differentiate the role of CEP85 in STIL stability from that of STIL recruitment is to check if overexpression of STIL can rescue centriole duplication defects in cells depleted of CEP85, i.e. completely bypassing the requirement of CEP85.

We thank the reviewer for her/his thoughtful comments. As suggested, we performed rescue experiments where we overexpress STIL in CEP85-depleted cells, to assess the level

of centriole duplication in S-phase. Our results indicate that expression of WT STIL as well as STIL L64A and R67A mutant are unable to rescue centriole duplication, supporting the role of CEP85 in facilitating STIL recruitment to centrioles (Supplementary Figure 9a-c). To further validate this point, we overexpressed a non-degradable form of WT STIL and STIL L64A and R67A mutant to assess their ability in centriole amplification¹. Consistently, we found that expression of similar level of STIL L64A and R67A mutant was unable to induce centriole overduplication to WT STIL levels (Supplementary Figure 9d-f). Together, our data support a dual role for CEP85 in STIL centriolar localization and its stability to control centriole duplication.

This comment from the reviewer also made us realize that it was necessary to investigate which factors are required for the recruitment of CEP85 to centrioles. To do this, we depleted CEP192, CEP152, PLK4 and STIL in U-2 OS cells and examined the impact on CEP85 centriolar localization. We found that depletion of CEP192, CEP152 and PLK4 led to a reduction in centriolar recruitment of CEP85 (Supplementary Figure 2f-g). These data suggest that CEP85 acts downstream of CEP192, CEP152, and PLK4, and therefore the model in Figure 7H has been modified accordingly. These results are mentioned on page 8 of the revised manuscript.

Reviewer #2:

The manuscript “The structural basis for CEP85-mediated control of centriole duplication” by Liu et al reports their identification of CEP85 as a novel centriole duplication factor directly interacting with the previously uncharacterized N-terminal domain of STIL to spatiotemporally regulate the early stages of centriole duplication. They first identified CEP85 as a new regulator of centriole duplication using BioID with several known centriolar proteins as the baits. Based on a series of RNAi and in vivo data, they found that CEP85 is required for robust accumulation of STIL at centrioles and PLK4 activation. The interaction between CEP85 cc4 and STIL NTD was mapped and confirmed by several biophysical techniques including in vitro pulldown, analytical ultracentrifugation, Y2H, and ITC. Based on the mapped binding sites, they determined the structures of CEP85 cc4 and STIL NTD, first individually to high resolutions of 2.1 and 1.7 Å respectively, and later as a complex to a low resolution of 4.6 Å. Based on the crystal structures, they identified critical residues mediating the interaction which was confirmed by mutagenesis analyses. Interaction-disrupting mutants of CEP cc4 were found to impair both centriole duplication and STIL recruitment to centrosomes, as well as prevent robust activation of PLK4 in vivo. Based on all these findings, they claim the elucidation of the molecular basis underlying a previously undescribed modulatory step during the most upstream events of centriole duplication. The article was written properly, and the reported finding is novel as they claimed. However, I found some of their data were not so convincing, particularly the complex structure which is the main finding of their work. Listed below are my major and minor concerns.

Major points:

1. Their ITC data show a strong and robust interaction between CEP85 cc4 and STIL NTD. However, in all their *in vivo* studies, the two proteins only partially (~15%) co-localized, which they concluded as a transient interaction. It is hard to understand how such a strong interaction *in vitro* leads to only a transient interaction *in vivo*? Does it imply there is another layer of regulation by an unknown mechanism?

The ITC/AUC experiments that we had presented in the manuscript suggested that the CEP85-STIL binding affinity is ~ 20 μM . Additional ITC and AUC experiments designed to clarify the binding stoichiometry of the complex (see comments below, Supplementary Figure 4c, Supplementary Figure 5g) suggest a K_D of ~ 60 μM . These values classify this interaction as a relatively weak interaction, which might partially explain the putatively transient interaction at centrosomes observed *in vivo*.

The complex might also play a role in the cytoplasm. Holland and colleagues propose that cytoplasmic STIL needs to associate with PLK4 to transform into a functional conformation in order to be recruited to centrioles². So, in analogy, we think that CEP85 may interact with both centrosomal and cytoplasmic pools of STIL to play its dual regulation on STIL. In agreement with this notion, our microtubule recruitment assay in Figure 6H-I indicate CEP85 can robustly recruit cytoplasmic STIL to microtubules. This is now also discussed in the manuscript. However, the reviewer is of course right in pointing out that there might indeed be further layers of regulation by unknown mechanisms that remain to be explored.

2. Their ITC data (Fig 4D) show “ $N=0.34$ ”, which is inconsistent with their tetrameric structural model of the complex. How to explain this? Similarly, the native MS data (Fig S4D) show that, despite the majority of the tetrameric complex, a substantial fraction of the complex are trimers (2xSTIL:1xCEP85) or even pentamers (3xSTIL:2xCEP85). Further, given the strong dimeric interaction of the coiled coils, it is hard to understand why all CEP85 cc4 are monomers (Fig S4D)? Also looking strange is that most STIL NTD are actually dimers. Overall, it seems very ambiguous how the two proteins behave individually, and whether the crystal structure they determined is really the physiological complex of them *in vivo*, even they may indeed interact in the cell, transiently as was claimed by the authors.

Concerning the ITC binding stoichiometry: We had demonstrated by MALS that the CEP85 cc4 is in fact partially unstable at room temperature (Supplementary Figure 5b). We have now also performed thermal melts which emphasizes this point further (Supplementary Figure 5f). This instability probably explains the lower binding stoichiometry observed at room temperature compared to our structure (Supplementary Figure 5g, shortly discussed in the corresponding figure legend). However, to address the

point of binding stoichiometry directly, we have now done ITC experiments at reduced temperature (10°C) to stabilise the CEP85 cc4 (Supplementary Figure 5g) and also redone our AUC experiments under conditions optimised to resolve the binding stoichiometry of the complex (Supplementary Figure 4c). Both experiments demonstrate a binding stoichiometry of 1:1 (2:2), which is in perfect agreement with our structural data (the AUC data resolved both 1:2 and 2:2 complexes). In further support of the idea that the stability of cc4 somewhat compromises the observed ITC stoichiometry at 25 °C, we have now also performed binding experiments by ITC at 25 °C using a longer construct of CEP85. This construct contains an additional coiled coil element (cc5) and is more stable than WT CEP85 cc4 alone as judged by thermal melts (Figure 1B for the reviewers' attention). With this construct we observed robust binding to STIL NTD of comparable affinity and binding stoichiometry of 1:1 (2:2), as observed for CEP85 cc4 at 10°C (Figure 1A for the reviewers' attention).

Second, concerning the apparent ambiguous behaviour of the proteins individually. These are only observed in native mass-spectrometry experiments, where the proteins are injected at high concentrations (due to the relatively low K_D) and run in vacuum (removing the solvation shell of water). Under milder conditions, in solution and at lower concentrations, such as MALS (Supplementary Figure 5a+b) at room temperature, the CEP85 cc4 is in monomer-dimer equilibrium and the STIL NTD is predominantly monomeric. To address this point further, we also have performed AUC at reduced temperature (10 °C) with the individual proteins and find this notion (CEP85 cc dimer, STIL NTD monomer) confirmed. Thus, we believe that the different behaviour of the individual proteins observed in native mass-spectrometry might be down to the experimental conditions. The native mass-spectrometry data nevertheless confirms complex formation and also confirms that the CEP85 cc4 binds only as a dimer to the STIL NTD. It is therefore in agreement with our conclusions concerning CEP85-STIL binding.

3. Another concern about the structure of the complex is that it is between T. adhaerens STIL NTD, which shares only 28% identity with human ortholog, and human CEP85 cc4. It would be helpful to check the binding of both proteins from the same organism, e.g. T. adhaerens STIL vs T. adhaerens CEP85, by ITC to confirm the interactions are comparable in both cases. Along the same line, crosslinking-MS analyses would help to further validate the interaction sites given the very low resolution of their complex structure.

We had shown that point mutations in the interface of our structure strongly compromise the binding between the human proteins (based on pull-downs and a microtubule-based recruitment assay *in vivo*) as well as between chicken STIL NTD and human CEP85 cc4 in ITC experiments with recombinant proteins. These experiments argue that, despite being obtained with *Trichoplax* STIL NTD and human CEP85 cc4, our structure is nevertheless relevant. To strengthen this point further we have now also used cross-linking MS analyses, as suggested by the reviewer. This experiment demonstrates the presence of a specific cross-link between chicken STIL NTD and human CEP85 cc4 for the WT, but not the

mutant proteins. This cross-link is in agreement with our structural model (Supplementary Figure 6a-d).

In addition, we have now also conducted ITC experiments with human STIL NTD vs human CEP85 cc4 and chicken STIL NTD vs chicken CEP85 cc4 that show binding of the WT, but not the mutant proteins (Figure 1C+D for the reviewers' attention). Fitting the WT data to obtain the K_D and binding stoichiometry of these intra-species ITC data was difficult because the parameters were poorly constrained by the fit under the experimental conditions that we could access. Both human and chicken interactions may have lower binding affinities and, in the case of human STIL NTD, we were unable to obtain sufficiently concentrated stocks without evidence of aggregation. In the case of the chicken STIL NTD – human Cep85 cc4 experiments we required >1 mM STIL NTD stock (>41 mg/ml) and we were unable to duplicate this with the human construct. The chicken interaction may also occur with a more complex binding mode, for example possibly involving the conserved patch 2 of STIL NTD, or may require further optimisation of solvent conditions since the chicken cc4 sequence contains two cysteine residues that are well placed in the parallel cc dimer to form disulphide linked material, despite the presence of DTT in the buffer. Although we are reluctant to fit these data quantitatively, it is clear that the proteins from both species bind to each other and that the same mutants in the conserved binding interface that disrupt the chicken – human interaction also prevent binding for the corresponding intra-species interactions. All of this is in complete agreement with our structural model.

We also attempted ITC experiments at 10 °C using *Trichoplax* STIL NTD vs *Trichoplax* CEP85 cc4 but without success. However, SEC-MALS and CD experiments showed that the *Trichoplax* CEP85 cc4 was already partially unfolded even at 4 °C and essentially monomeric at room temperature (Figure 1E+F for the reviewers' attention). Since CEP85 cc4 needs to be dimeric to bind to STIL NTD (Supplementary Figure 4d), this instability compared to human (Supplementary Figure 5b, Supplementary Figure 5f) and chicken CEP85 cc4 (Figure 1F for the reviewers' attention) likely explains the lack of a robust interaction in ITC experiments. CEP85 contains a number of additional coiled coil domains (see Figure 4A for a domain overview of CEP85) that might act to stabilise cc4 dimer formation in the context of full length CEP85. Indeed, using human CEP85, we found that in the absence of cc4, CEP85 is still able to oligomerise (Figure 1G for the reviewers' attention).

4. Regarding the unwinding of the N-terminal part of CEP85 cc4 upon binding of STIL NTD, more discussion of the cause and potential function should be considered. Could this be tested by some techniques such as limited proteolysis or NMR? Does a short version of CEP85 cc4 corresponding to the seen part in the complex structure bind in the same manner (and with a similar affinity) as the one used in their studies? If so, SAXS might be used to further confirm the tetrameric complex.

We had tried in the past to shorten the CEP85 cc4 further, but did not see any binding to STIL NTD. This is not unexpected, since CEP85 cc4 can only bind as a dimer to STIL (Supplementary Figure 4d) and taking off more heptad repeats from the coiled coil would act to destabilise dimer formation (the coiled coil is already partly unstable at room temperature (Supplementary Figure 5b+f)).

In our opinion, it would be very challenging to use limited proteolysis to check the fraying of the CEP85 cc4 in solution and when bound to STIL. The binding affinities are relatively weak making it difficult to obtain unique complexes and therefore the nature of the relevant controls is not clear. Thus, any result would be difficult to interpret unambiguously. SAXS under the required high protein concentrations (relatively low K_D) would also lead to technical difficulties (concentration effects, multiple states, aggregation etc.), besides being unlikely to be able to differentiate between a fully folded or partially unfolded N-terminal coiled-coil part in the complex (SAXS is a low resolution technique). NMR experiments would require a full assignment of the CEP85 cc4 dimer, which is beyond the time-scale of this revision (and, due to the nature of the parallel coiled coil dimer, being non-trivial). Thus, unfortunately, we feel that we are unable to address this point experimentally. However, partial unwinding of proteins to enable crystal packing is not uncommon in protein crystallography. Thus, as discussed in the manuscript, we believe that this is the most likely explanation in our case as well, especially given the partial instability of the CEP85 cc4 at room temperature.

Minor points:

1. “Conservation Scale” bars were shown several times (Fig 5A, Fig S3C, Fig S7D). However, it was unclear how the values were calculated? Further, more homologous proteins should be included in their alignments (Fig S7B&C) to demonstrate residue conservations.

The conservation scores for both CEP85 cc4 and STIL NTD, were calculated with ConSurf using manually refined multiple sequence alignments each of which contained 136 non-redundant homologous sequences from the same set of species. The position-specific scores were calculated using a Bayesian algorithm. These scores are divided into a discrete scale of nine grades and indicate the relative degree of evolutionary conservation at each amino acid position in the given alignment. We integrated this information into the Materials and Methods.

For an extended multiple sequence alignment please refer to Figure 2 for the reviewers’ attention. The extended alignment for both CEP85 cc4 and STIL NTD includes sequences from diverse metazoan organisms representing the main branches of the phylogenetic tree shown in Supplementary Figure 8. It clearly shows that the residues mutated in our study are highly conserved across species. Similarly, the regions (blocks) corresponding to the secondary structural elements are well conserved. The number of Supplementary Figures is limited and these alignments are very bulky without, to our mind, carrying extra or

essential information. Thus, we would prefer to retain in the manuscript supplement their shorter version that includes the organisms used in our study.

2. *How to explain that depletion of CEP85 reduced STIL level, but depletion of STIL rather increased CEP85 level (Fig 2D)?*

This is a very good question and we must confess that this is also a puzzling observation for which we have no concrete explanation. Our data clearly indicate that CEP85-STIL can form a complex *in vitro*, and we think a plausible explanation is that the regulated stability of this complex (or its individual components) *in vivo* may be a significant factor determining its bioavailability. We have shown that CEP85 acts upstream of STIL in centriole duplication. In the absence of CEP85, STIL is unable to fulfill its physiological functions and therefore may promote its degradation by specific E3 ubiquitin ligases. This is not without precedent since other centriole duplication factors^{1,3-6} (SASS6, CPAP) have been shown to regulate cellular levels of core duplication factors. Consistent with this observation in Figure 2D, our new data indicate that STIL depletion also increased the centriolar level of CEP85 (Supplementary Figure 2f-g). Those data imply a potential feedback regulation of CEP85 levels that warrants further investigation. We previously reported a similar phenomenon that depletion of CEP120, SPICE1, CPAP and CEP135 resulted in a marked increase in the PLK4 signal surrounding the mother centriole⁷. Further work is needed to identify UPS components that potentially regulate the CEP85/STIL complex.

3. *Regarding the reasoning of absence of CEP85 in nematodes and flies (Fig S7A), it would be helpful to compare the structure of the STIL NTD reported here with that of the counterpart in Sas5/Ana2.*

Intriguingly, the fly and nematode homologs of STIL, Ana2 and Sas5, both do not have a NTD (schematically shown in Supplementary Figure 8). In fact, our sequence analysis revealed that insects such as wasps, ants, butterflies, beetles and bees also lack the STIL NTD. We were unable to identify a CEP85 homologue in these organisms, which could be taken as a further indication that the described interaction between CEP85 and STIL is evolutionarily relevant. A small paragraph in the discussion of the manuscript describes our findings (also see Supplementary Figure 8a). As discussed in the legend of Supplementary Figure 8, conclusions concerning nematode SAS-5 are difficult though, since its homology to STIL is not apparent from sequence comparison.

Reviewer #3

The manuscript by Liu et al describes the identification the protein CEP85 as a new player in centriole duplication. Initially, the protein CEP85 is identified to be interacting with known centriole duplication factors by a set of protein proximity detection methods, and subsequently, its role in the centriole duplication process is established and validated in vivo. Two high resolution structures, one of the interacting domains CEP85-CC4 and one of its binding partner STIL-NTD are determined, and a low resolution structure of the protein-protein complex is presented that fits well to the evolutionary conservation of these domain. The interaction between CEP85 and STIL is further characterized by biophysical methods including NMR spectroscopy. Single-point mutations, based on the crystal structures are analyzed in vitro and in vivo, demonstrating that the CEP85-STIL interaction is essential for STIL localization to centrioles and subsequent PLK4 activation and finally correct daughter centriole formation. The work comprises an overall impressive amount of data and experiments, which appear all very well done and well documented. The amount of work shown here is clearly above average and fully sufficient to warrant publication. The manuscript is clearly written and the provided findings represent a major scientific step forward in the centriole field. I highly recommend publication in Nature communications. Two minor issues may be addressed:

We thank this reviewer for her/his enthusiasm.

1.) The title "Structural basis of .." is in my view even too modest and includes only part of the achievements that the manuscript provides. Perhaps the authors want to change the title to a more general term to highlight that they also identify the role of CEP85 in the first place and elucidate its function in vivo at least partially.

We appreciate this kind suggestion. We changed the title to “Direct binding of CEP85 to STIL ensures robust PLK4 activation and efficient centriole assembly”.

2.) The authors conclude from the absence of a detectable interaction with the R67A mutant in NMR and ITC experiments (Fig. S3), that CEP85 does not interact with STIL via its patch2. This conclusion should be worded and discussed more carefully. The only statement that can be made safely is that under the conditions used in vitro, this interaction not detected. It may nonetheless exist in vivo, perhaps with a phosphorylation or other regulation or under otherwise different conditions.

This is a good point. We now mentioned on Page 22 of the manuscript “While the other patch would be well placed to contribute to CEP85 binding (Supplementary Figure 3c), NMR experiments suggest that this is not the case at least under the condition used in vitro (Supplementary Figure 3d)”

REVIEWERS' COMMENTS:

Reviewer #1 (Remarks to the author):

The authors have fully addressed the issues I raised. The story is interesting and important and I therefore fully support its publication in nature communication.

Reviewer #2 (Remarks to the author):

My comments and concerns have been adequately addressed. In my opinion, it is ready for publication.

Responses to the reviewers

Reviewer #1

The authors have fully addressed the issues I raised. The story is interesting and important and I therefore fully support its publication in nature communication.

We thank the referee to support publication in Nature Communications

Reviewer #2

My comments and concerns have been adequately addressed. In my opinion, it is ready for publication.

We thank the referee to support publication in Nature Communications

References

1. Arquint, C. & Nigg, Erich A. STIL microcephaly mutations interfere with APC/C-mediated degradation and cause centriole amplification. *Current Biology* **24**, 351-360 (2014).
2. Moyer, T.C., Clutario, K.M., Lambrus, B.G., Daggubati, V. & Holland, A.J. Binding of STIL to Plk4 activates kinase activity to promote centriole assembly. *The Journal of Cell Biology* **209**, 863-878 (2015).
3. Puklowski, A. *et al.* The SCF–FBXW5 E3-ubiquitin ligase is regulated by PLK4 and targets HsSAS-6 to control centrosome duplication. *Nature Cell Biology* **13**, 1004 (2011).
4. Rogers, G.C., Rusan, N.M., Roberts, D.M., Peifer, M. & Rogers, S.L. The SCF Slimb ubiquitin ligase regulates Plk4/Sak levels to block centriole reduplication. *The Journal of Cell Biology* **184**, 225-239 (2009).
5. Cunha-Ferreira, I. *et al.* The SCF/Slimb Ubiquitin Ligase Limits Centrosome Amplification through Degradation of SAK/PLK4. *Current Biology* **19**, 43-49 (2009).
6. Kim, M.K., Dudognon, C. & Smith, S. Tankyrase 1 regulates centrosome function by controlling CPAP stability. *EMBO Reports* **13**, 724-732 (2012).
7. Comartin, D. *et al.* CEP120 and SPICE1 cooperate with CPAP in centriole elongation. *Current Biology* **23**, 1360-1366 (2013).

Figure 1 for reviewers

Figure 1 for reviewers

(A, B). Recombinant chicken STIL NTD and the human C-terminal CEP85 construct CEP85⁵⁶²⁻⁷⁶² (including CEP85 cc4 and cc5) directly interact with each other at 25 °C. Binding affinity and stoichiometry are comparable to ITC experiments with human CEP85 cc4 at 10 °C (Supplementary Figure 5g). At 10 °C cc4 dimer formation is stabilised (Supplementary Figure 5f). (A). ITC binding isotherm for chicken STIL NTD titrated into human CEP85⁵⁶²⁻⁷⁶² at 25 °C. The resulting K_D , ΔH and STIL NTD/CEP85⁵⁶²⁻⁷⁶² binding stoichiometry (N) as an average from a total of three measurements are indicated. (B). CD-based thermal melting analysis of recombinant human CEP85⁵⁷⁰⁻⁶⁶² cc4 and CEP85⁵⁶²⁻⁷⁶². Note the increased thermal stability of CEP85⁵⁶²⁻⁷⁶² compared to CEP85⁵⁷⁰⁻⁶⁶² cc4. CEP85⁵⁶²⁻⁷⁶² contains an additional coiled coil domain (cc5, Figure 4A). (C, D). Recombinant STIL NTD and CEP85 cc4 from different species directly interact with each through a conserved interface. (C). ITC binding isotherm for human STIL NTD (WT and R67A mutant) titrated into human CEP85 cc4 (WT and Q640A) at 10 °C. (D). ITC binding isotherm for chicken STIL NTD (WT and R63A mutant (R67A in human STIL)) titrated into chicken CEP85 cc4 (WT and Q659A mutant (Q640A in human CEP85)) at 10°C. At this temperature, these interactions are driven by a large favourable entropy of binding similar to what is observed with the chicken STIL NTD – hs CEP85 cc4 titrations (Supplementary Figure 5g). Please note that it was not possible to fit the data from (C, D), unambiguously to obtain K_D and binding stoichiometries, due to the relevant parameters being poorly constrained by the fit. This could be due to several reasons, e.g., in the case of the human STIL NTD, an aggregation tendency, somewhat lower binding affinities, or more complex binding modes that might involve additional interaction sites (e.g. conserved patch 2 of STIL NTD). (E, F). Recombinant *Trichoplax* CEP85 cc4 does not form stable dimers in solution. (E). Size exclusion chromatography - multi-angle light scattering (SEC-MALS) chromatograms of recombinant *Trichoplax* CEP85 cc4 run at room temperature at varying concentrations. Shown are the respective refractive index signals together with the derived molar masses (indicated by thicker horizontal lines). The calculated, theoretical molecular weight is indicated. *Trichoplax* CEP85 cc4 remained monomeric over the concentration range examined (the average molecular weight in the indicated regions ranged from 13-15 kDa). (F). CD-based thermal melting analysis of recombinant *Trichoplax* and *Gallus* CEP85 cc4 both at 0.6 mg/ml. Note the low thermal stability of the *Trichoplax* coiled coil. (G). CEP85 contains additional oligomerisation domains besides its cc4 domain. 293T cells expressing Tet-inducible FLAG-BirA* or FLAG-BirA* tagged human CEP85 WT transgenes were transfected with MYC-CEP85 WT or cc4 deletion constructs for 48 h in the presence of tetracycline (2µg/mL), and immunoprecipitated using FLAG antibody-conjugated beads. FLAG-BirA* CEP85 and co-immunoprecipitated proteins were probed with the indicated antibodies.

Figure 2 for reviewers

The STIL NTD/CEP85 interaction interface is conserved across metazoans. Extended multiple sequence alignment of STIL NTD (A) or CEP85 cc4 (B) homologues from diverse metazoan organisms, representing the major branches of the tree shown in Supplementary Figure 8. The alignments are coloured according to the CLUSTAL coloring scheme, residue color intensity is based on conservation. The secondary

structure elements are shown above the alignment. The two STIL / CEP85 residues mutated in this study are indicated with red dots.